# Ocean driven retreat of the Northeast Greenland Ice Stream following the Last Glacial Maximum

S. Louise Callard [1] ✉, Colm Ó Cofaigh[2], Jeremy M. Lloyd [2], James A. Smith [3], Catalina, A. Gebhardt [4], Torsten Kanzow [4,5] & David H. Roberts[2]

The Northeast Greenland Ice Stream (NEGIS), the largest ice stream draining the Greenland Ice Sheet (GrIS), is losing mass at an accelerating rate due to atmospheric and ocean-driven melting. Holding the equivalent of 1.1–1.4 metres of sea-level rise, its collapse will have a significant impact on global sea levels making it crucial to understand the controls on its dynamic behaviour. The NEGIS retreated from the continental shelf edge by 21.6 ka BP, with this study confirming continued grounding line retreat to ~100 km from the shelf edge by 20.3 ka BP, earlier than previously reported. This early retreat was driven by warm Return Atlantic Water (RAW) and amplified by a retrograde seabed, which together drove initial grounding line retreat. The presence of a series of grounding zone wedges indicates a quasi-stable grounding line, which was fronted by an ice shelf. Grounding-line retreat took place between 20.3 and 15.2 ka BP. However, ice-shelf break up caused by enhanced sub-ice shelf melt rates driven by RAW ingression, coupled with surface thinning instigated by atmospheric warming during Greenland Interstadial I, triggered rapid retreat of the ice stream after 15.2 ka BP. Our findings confirm the dominant role of oceanic forcing in grounding line stability and ice-shelf break up.

The greatest uncertainty to future sea-level projections, yet potentially the largest contribution, stems from the uncertain response of the Antarctic and Greenland ice sheets to future warming[1,2]. The Northeast Greenland Ice Stream (NEGIS) extends more than 600 km into the interior and drains approximately 12% of the Greenland Ice Sheet, (GrIS). Complete collapse of the NEGIS would raise global sea level by 1.1–1.4 m[3,4]. The NEGIS is fronted by three marine-terminating glaciers: Nioghalvfjerdsfjorden Glacier (79N), Zachariae Isstrøm (ZI) and Storstrømmen Glacier (SG). These fringing glaciers and ice shelves were relatively stable up until the mid-2000s, but since then, 79N and ZI have shown significant flow acceleration and dynamic thinning[5] with complete loss of the ice shelf fronting ZI in late 2012 and early 2013 (up to 40 km loss[3]). 79N still retains its 70 km long, 20 km wide ice shelf, which buttresses the ice stream. However, a significant calving event observed in 2022 indicates ongoing ice shelf instability, with models predicting that the collapse of the ice shelf would increase grounding line flux by over 160%[6]. Enhanced atmospheric- and particularly ocean-driven melting has been identified as the primary driver of retreat[3–5,7,8]. Numerous studies have indicated that Return Atlantic Water (RAW, also referred to in the literature as Return Atlantic Current, RAC) is reaching the grounding line of 79N and driving both grounding line and sub-ice shelf melt[7,9–12] (Fig. 1b).

Given these recent changes, it is increasingly important to understand the future evolution of the NEGIS in order to assess future sea-level rise. While numerical simulations provide a powerful predictive tool, their reliability depends on an accurate parameterisation of ice-sheet dynamics and key boundary conditions such as bathymetry and climate forcing. This issue is particularly acute at the ice

[1]Department of Geography, University of Newcastle, Newcastle, UK. [2]Department of Geography, Durham University, Durham, UK. [3]British Antarctic Survey, Cambridge, UK. [4]Alfred Wegener Institute Helmholtz Centre for Polar and Marine Research, Bremerhaven, Germany. [5]Institute of Environmental Physics, University of Bremen, Bremen, Germany. ✉e-mail: louise.callard@ncl.ac.uk

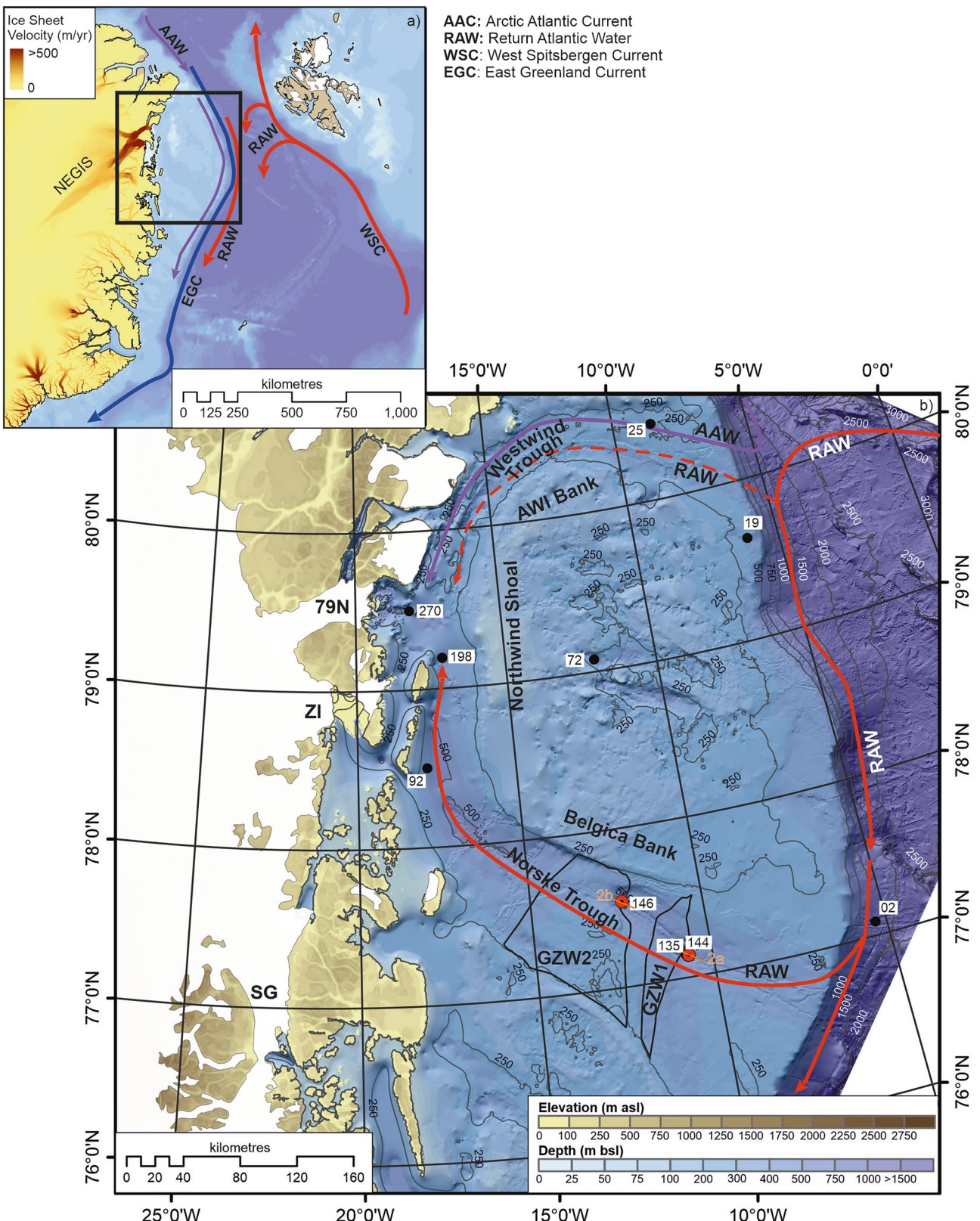

AAC: Arctic Atlantic Current
RAW: Return Atlantic Water
WSC: West Spitsbergen Current
EGC: East Greenland Current

**Fig. 1 | Location map of the study site. a** Overview of the study area with surface ocean circulation pattern: colours denote relative warmth of the currents, with red being the warmest and blue coldest. The ocean bathymetry is from GEBCO[57]. Ice sheet velocity provided by the NASA MEaSUREs ITS_LIVE project[58] and the Northeast Greenland Ice Stream (NEGIS) is labelled on the map. **b** Close up of the northeast Greenland continental shelf with cores discussed in the text: 144, PS100-144GC and 146, PS100-146GC (red; this study), 270, PS100-270[24]; 198, PS100-198[27]; 92, core DA17-092G[26]; 72, core DA17-72G[25]; 25, core PS93-025[59]; 19, DA17-019G[17]; 135, Core DA17-NG-ST12-135G[22]; 02, core GR02-GC[19]. The location of the two sub-bottom profiler lines displayed in Fig. 2 is shown in orange. Position of the three separate glaciers, Nioghalvfjerdsfjorden Glacier (79N), Zachariae Isstrøm (ZI), and Storstrømmen Glacier (SG) forming the NEGIS is indicated in black. Estimated position of Grounding Zone Wedge (GZW)1 and GZW2 from López-Quirós et al.[22].

sheet grounding line, where there is often a dearth of reliable bathymetric data or information on ocean heat transport. Alternatively, the palaeo record provides an unparalleled archive of past changes, enabling an understanding of ice-ocean interactions, and particularly the threshold of ice sheet variation in response to thermal drivers, bathymetry and internal ice-sheet dynamics. This palaeo perspective is essential to assess the significance of recent changes, to provide empirical evidence to refine models of change, and to gain an understanding of the overall stability of the NEGIS system[13].

### Background to study site

The continental shelf of northeast Greenland is up to 300 km wide and is dissected by two arc-shaped troughs: the Westwind Trough in the north and the Norske Trough in the south. These are separated by the shallower AWI Bank, Northwind Shoal and Belgica Bank, and converge in the embayment directly fronting the 79N ice shelf (Fig. 1). The 350 km long Norske Trough is widest at the shelf edge, reaching a width of over 200 km, and narrows to ~ 50 km on the inner shelf. The trough is characterised by a series of deep basins reaching between 400–500 m below sea level, separated by bathymetric highs. The deeper Norske Trough provides a crucial route for RAW to the inner shelf and the grounding line of 79N[10,12,14].

Recent evidence from marine cores, bathymetric and geophysical observations from the continental shelf, provides a longer-term reconstruction of the behaviour of the NEGIS. The maximum extent of the NEGIS during the Last Glacial Maximum (LGM) has been debated[15–17], but bathymetric and geophysical observations record glacial lineations and grounding zone wedges (GZWs) up to the shelf edge[16,18]. Additionally, glaciogenic debris flows on the continental slope in front of Norske Trough provide indirect support for the extension of fast-flowing, grounded ice to the shelf edge during the LGM[18]. The timing of the NEGIS reaching the shelf edge is currently not constrained. However, evidence from core material dated on the continental slope indicates maximum extent was likely ~23.5 ka BP[19], with the earliest age of initial retreat from the outer shelf being 21.6 ka BP[18]. This is consistent with recent evidence for ice-sheet thinning through the coastal mountains of northeast Greenland at ~22ka[13]. Elevated concentrations of ice-rafted debris (IRD) and terrigenous input recorded in cores on the continental slope between 21 and 16 ka BP also point towards an active outer-shelf ice stream during this period[20,21].

Multibeam bathymetric and sub-bottom profiler data record a range of glacial landforms that record the dynamic retreat of the NEGIS from its maximum extent[16,22]. A mixture of recessional moraines and GZWs mapped in both Westwind and Norske troughs highlights a varied style of retreat during the early deglacial period[16,18,22,23]. In the Norske Trough, a series of moraine ridges on the outer to middle trough indicate a stepwise retreat pattern[22] (Fig. 1b), while two large GZWs (GZW1 and GZW2[22]; Fig. 1b) in the mid-trough indicate periods of longer stability.

A series of core records published from the inner Norske Trough (Fig. 1b) confirms the importance of oceanic warming and the role of RAW on the retreat of the grounding line since 13 ka BP[24–27]. Modelled simulations (constrained by marine sediment records) of the Early Holocene evolution of the GrIS show the relationship between grounding line retreat across the inner shelf and ice surface elevation lowering[28]. However, to date, there is limited information on the role of the ocean during early deglaciation and, therefore, on the drivers of the initial retreat of the NEGIS from its maximum extent. Palaeoceanographic constraint for models of past NEGIS behaviour are limited predominantly to the Holocene. Here we present sedimentary and biostratigraphical data for the past 20 ka BP that constrains the timing of the initial retreat of the NEGIS and critically provides evidence of the controls on grounding line stability and ice-shelf break up (Fig. 1). Core PS100-144GC (hereafter 144GC) lies between two GZW's (a previously

unidentified GZW and GZW1; Figs. 1 and 2). Core PS100-146GC (hereafter 146GC), is taken directly seaward of GZW2 (Figs. 1 and 2). The surface of GZW2 contains a series of smaller ridges likely reflecting marginal retreat across GZW2 (Fig. 2) and infers quasi-stability during initial ice withdrawal from the outer to mid Norske Trough.

## Results

### Early deglacial and Atlantic water influence (>20.3 ka BP–17.6 ka BP)

The sediments obtained from cores 144GC and 146GC provide evidence for ice-stream activity during deglaciation from the end of the LGM in the outer Norske Trough. Four lithofacies (LF1-LF4) represent subglacial conditions during ice advance, followed by glaciomarine and sub-ice shelf conditions during retreat (Fig. 3). Foraminiferal analysis conducted on core 144GC supports the lithofacies interpretation with the assemblage being split into five foraminiferal assemblage zones (FAZ's) (Fig. 4).

Lithofacies 1 (LF1) is the basal lithofacies in both cores. It is characterised by a matrix-supported, consolidated dark-grey diamicton with variable grain size, absence of foraminifera (FAZ1), high wet bulk density, medium magnetic susceptibility, and relatively high shear strength (11–32 kPa, Figs. 3 and 4). Sub-bottom profile data (Fig. 2) reveal this unit to be streamlined and ubiquitous across the seafloor. It is interpreted as a subglacial traction till[15], an interpretation supported by López-Quirós et al.[22] at other adjacent sites in the outer Norske Trough (Fig. 1). Core 146GC contains a duplication of LF1 in the lower stratigraphy, suggesting grounding line oscillation and the deposition of a re-advance till (Fig. 3). In contrast, based on its upper stratigraphy (sub-ice shelf to glaciomarine transition), core 144GC records the grounding line migration west of the core site with no re-advance over the site. The geomorphic and chronological evidence from other studies indicates that the outer Norske Trough was deglaciated by 21.6 ka BP[18].

Lithofacies 2 (LF2) consists of a finely laminated to crudely stratified silt clay and is colour banded (Fig. 3). It drapes LF1 and infills lows in the seafloor topography (Fig. 2). Shear strength is low (<10 kPa). This lithofacies is largely devoid of gravel/pebble-sized clasts and is faulted in places. It is characterised by medium-low wet bulk density, medium magnetic susceptibility values and low shear strength (Fig. 3). The benthic foraminiferal assemblage in LF2 can be split into three assemblage sub-zones (Fig. 4). FAZ2a is dominated by *Stetsonia horvathi* (50–90%) with abundance increasing to a maximum of 90% at the top of this unit. *Stainforthia. feylingi* (4–14%), *Cassidulina neoteretis* (0–22%) and *Cassidulina reniforme* (1–14%) are common but decreasing in abundance up core. Benthic foraminiferal abundance through this zone is relatively low (foraminiferal accumulation rate (FAR) under 3000 indiv. cm$^{-2}$ kyr$^{-1}$) except for the lowermost sample where FAR exceeds 31,000 indiv. cm$^{-2}$ kyr$^{-1}$ (Fig. 4). Planktic foraminifera are also present in low concentrations (<500 indiv. cm$^{-2}$ kyr$^{-1}$), although again exceeding 2500 indiv. cm$^{-2}$ kyr$^{-1}$ in the lower-most sample (Fig. 4). The planktic $\delta^{18}O$ record in 144GC records low values through this zone (-2‰) except for an isolated spike 175 cm (Fig. 4).

These sediments are interpreted as proximal glaciomarine plumites[24,29]. A radiocarbon date of 18.6 ka BP towards the base of LF2 (15 cm above the subglacial till) provides a minimum age for deglaciation of the site of 144GC and, hence, retreat from the shelf edge. The age-depth model provides a date of 20.3 ka BP for the transition from subglacial till to glaciomarine sedimentation (see Supplementary Information 3 and 4). Core 144GC, which is located ~100 km from the continental shelf edge, supports continued early deglaciation and retreat across the outer continental shelf during this early deglacial period. This date is significantly earlier than published chronologies at a similar core location (16.6 ka BP in core 135, Fig. 1) and takes place during the global LGM[30]. While these cores cannot constrain whether

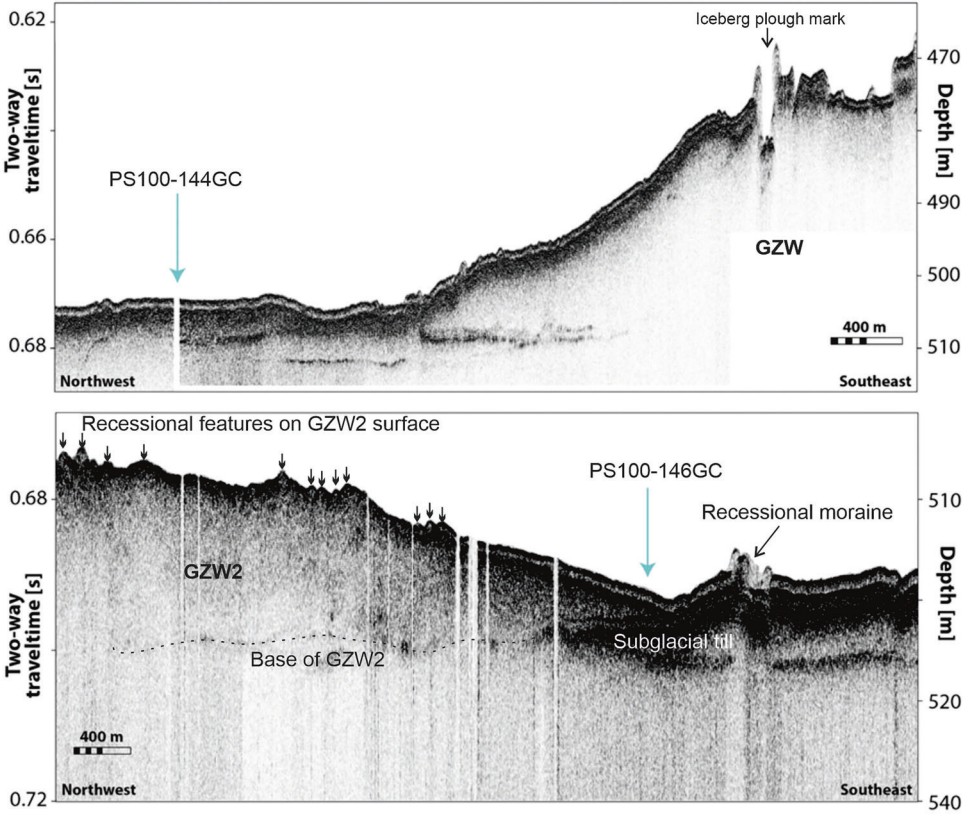

**Fig. 2 | Sub-bottom profiler data across the the two core locations 144GC and 146GC.** Data was collected using a hull-mounted Parasound DS III-P70 system with noticeable Grounding Zone Wedges (GZWs) and moraines labelled.

ice was grounded at the shelf edge, geomorphic evidence consisting of mega-scale glacial lineations (MSGLs), GZWs, and glaciogenic debris flows on the continental slope all indicate ice extended to the shelf edge[16,18], with radiocarbon dates indicating ice retreat from the shelf edge underway by 21.6 cal ka BP[18]. The age of the first occurrence of LF2 in core 146GC is not well constrained, but its presence between two till layers indicates that glacimarine conditions occurred between the periods of ice grounding at the site.

The benthic foraminiferal assemblage (FAZ2a) within core 144GC indicates harsh surface water conditions indicative of an ice shelf, with a dominance of *S. horvathi* and a secondary assemblage of *S. feylingi*[31,32](Fig. 4). These conditions persist throughout FAZ2a until *c.* 17.6 ka cal BP. An ice-shelf interpretation is further supported by the fine-grained nature of this unit and lack of IRD (Fig. 3), implying the presence of an ice shelf precluded the movement of icebergs over the core site (c.f.[29]). However, the occurrence of *C. neoteretis* is evidence of RAW presence immediately on deglaciation[33], the influence of which decreases up core. This indicates that RAW was present in the trough at depth during this early deglacial phase. This is consistent with observations from core GR02-GC[19], which records the presence of *C. neoteretis* from 35 ka BP to present at the continental shelf edge. Where the two records differ is the strength of the RAW signal, which is more variable on the continental shelf (this study) compared to the continental slope record[19]. The presence of *C. reniforme*, a glacimarine species found at present on the northeast Greenland continental shelf, is associated with chilled Atlantic water[32,34]. It covaries with *C. neoteretis* and provides further support for an Atlantic water presence during initial deglaciation. The occurrence of planktic foraminifera species, albeit in low concentrations, within FAZ2a does indicate open water conditions nearby, which suggests these species were advected to the core site under an ice shelf. Furthermore, the planktic $\delta^{18}O$ record in

144GC shows low values, indicating there was a significant meltwater signal during this early deglacial period.

## Grounding line stability and continued ice-shelf presence (17.6 ka–15.2 ka BP)

The early deglaciation from the continental shelf edge was followed by slow retreat and subsequent grounding line stabilisation as indicated in shelf bathymetry, geophysical data[22] and the sediment cores reported here (144GC and 146GC). From 17.6 ka BP to 15.2 ka BP, both cores consist of fine-grained, laminated sediments containing no IRD (LF2). The foraminiferal assemblage transitions from FAZ2a to FAZ2b and is characterised by very low benthic foraminiferal abundances (ranging between 16–2522 indiv. cm⁻² kyr−1), with foraminiferal numbers being too low (3–25 individuals counted) for a statistically robust count. However, although in very small numbers, *S. horvathi* and *S. feylingi* are most abundant, with occasional occurrences of *C. neoteretis* and *C. reniforme*.

FAZ2c is characterised by a significant increase in benthic foraminiferal concentration and is dominated by *S. horvathi* (46–53%) with *S. feylingi* also being common (4–32%). The abundance of *C. neoteretis* (3–13%) also increases in this section of the core, but *C. reniforme* is present in low abundance (2–4%), and planktic foraminifera remain absent. Both 144GC and 146GC continue to reflect sedimentation in an ice-proximal glaciomarine environment derived from meltwater plumes[29]. The lack of large clasts in the X-radiographs and fine particle size distribution (containing no sand component, Supplementary Fig. 1) indicates ice-berg rafting was suppressed (Fig. 3). Indeed, when combined with the very low abundances of benthic foraminifera and the absence of planktic foraminifera, this suggests the continued presence of an ice shelf at the time of deposition[35,36].

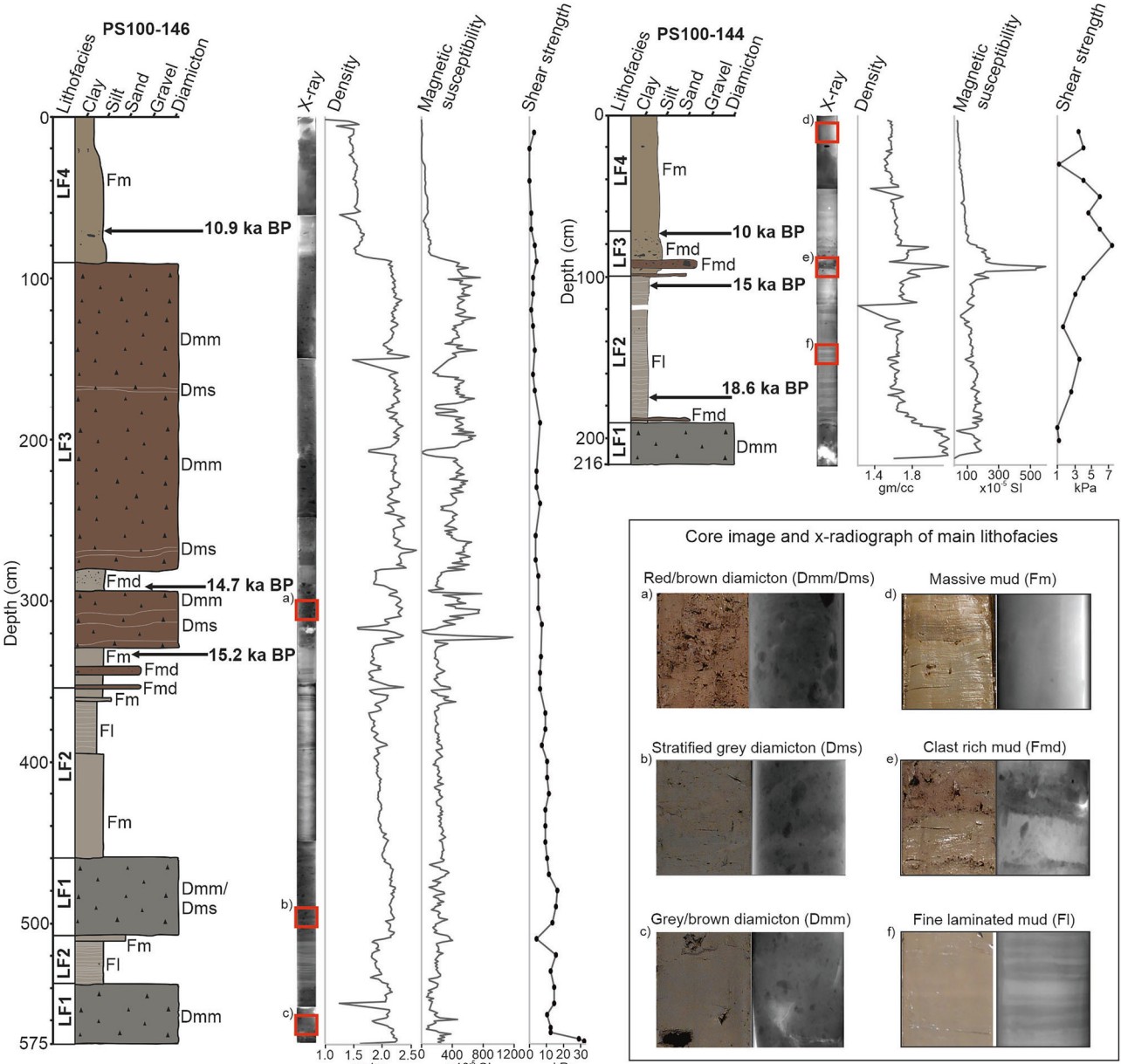

**Fig. 3 | Core logs and physical properties of 146GC and 144GC plotted against depth.** Core image and X-radiograph (X-ray) of the main lithofacies are provided in (**a-f**) and are marked on the core X-ray images in the red boxes.

Lithofacies codes used in the figure include matrix-supported diamicton that is massive (Dmm) or stratified (Dms) and mud that is massive (Fm), massive with dropstones (Fmd) or laminated (Fl).

Alternative explanations for the low abundance or absent foraminiferal counts include high sedimentation rates and/or poor preservation. However, there is no clear evidence of a change in sedimentation rate that could reduce benthic foraminiferal productivity[37]. We also rule out poor preservation as there are no agglutinated foraminifera or test linings present in this section of the core (evidence of poor preservation[37]), and the occasional calcareous species that are present are well preserved. Therefore, the presence of an ice shelf is the most likely explanation for the low species concentrations[(c.f.31)]. The substantial reduction in foraminiferal concentrations in FAZ2b and the lack of any planktic specimens may indicate the distance of the core site to the ice-shelf margin increased and, thus, inhibited the advection of foraminifera to the core site.

FAZ2c is characterised by an increase in benthic foraminiferal concentrations, dominated by the ice shelf and indicator species. However, *S. horvathi* decreases in abundance up core whilst *S. feylingi* and the RAW indicator species *C. neoteretis* and *C. reniforme* increase in abundance. The decrease in *S. horvathi* up core and the increase in *S. feylingi* suggest the ice shelf may have retreated, with the site becoming more proximal to the ice-shelf margin. The increase in *C. neoteretis* and *C. reniforme* indicates an increase in RAW ingress potentially due to the edge of the ice shelf retreating (calving) closer to the core site, leading to more favourable light and nutrient conditions[38]. However, the continued absence of planktic foraminifera, combined with the lack of IRD, indicates that the ice shelf was still present over the core site during this period.

It should be noted that an alternative hypothesis to an ice shelf is the presence of sea-ice, and *S. feylingi* is often associated with sea-ice edge/seasonal sea-ice conditions[38,39]. However, ice-shelf presence is reinforced by the geomorphological record within Norske Trough[16,22]. GZWs are considered to be a critical, diagnostic feature of ice streams that terminate into floating ice shelves[40]. Furthermore,

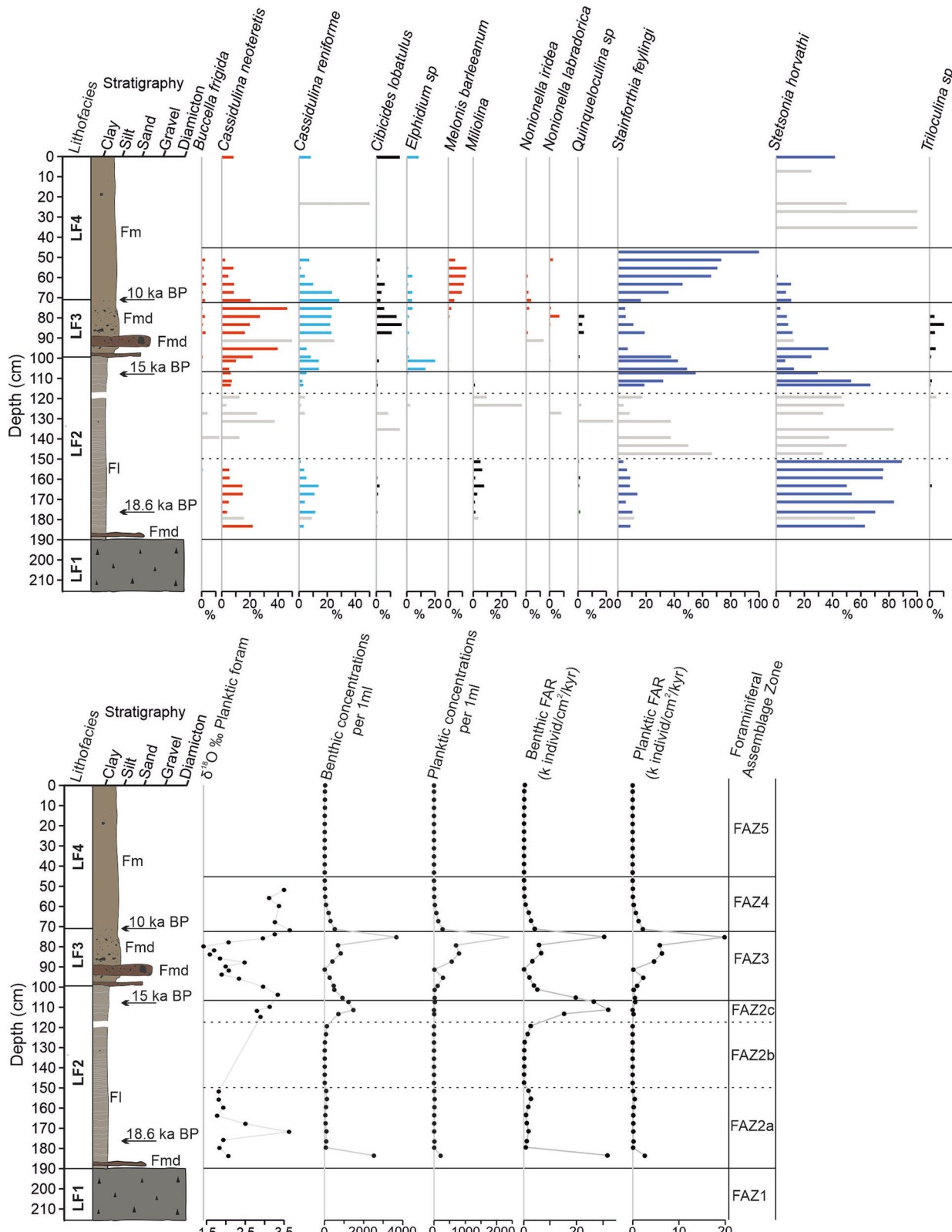

**Fig. 4 | Foraminiferal results for core 144GC.** Results include foraminiferal assemblage as relative abundance (species >5% shown), δ¹⁸O record of planktic foraminifera, benthic and planktic concentration record and foraminiferal accumulation rate (FAR) for benthic and planktic species for core 144GC. Red shaded species: Atlantic Water indicators; light blue shaded species: Polar water/ glacimarine indicators; dark blue shaded species: sea-ice indicators; grey shaded species: horizons with very low (<25) foraminiferal specimen counts. Lithofacies codes used in the figure include matrix-supported diamicton that is massive (Dmm) and mud that is massive (Fm), massive with dropstones (Fmd) and laminated (Fl).

there is evidence of tabular iceberg keel marks on the margins of the Norske Trough in front of GZW1 and GZW2 and near core 144GC[16,18,22]. It is likely that these icebergs calved from floating ice shelves[41,42]. The keel marks are noted at <300 m water depth[16] indicating a thick ice shelf fronting the grounding line at GZW1 and GZW2. The maximum extent of the ice shelf is unknown. However, if the NEGIS was pinned on GZW2, it would have extended at least 60 km to cover both 146GC and 144GC core locations. The presence of a floating ice shelf in the Norske trough is also supported by foraminiferal assemblages, including the presence of the ice-shelf indicator species *Glomulina oculus*, and laminated lithofacies in core 92G[26], which shows that an ice shelf fronted the NEGIS as it retreated along Norske Trough from the outer to inner shelf.

Together, cores 144GC and 146GC indicate that the grounding line receded to the mid-trough GZW2 between 20.3 and 15.2 ka, where it stabilised. This is supported by sedimentological variations in core 146GC, which contains two dark grey diamicton units separated by finely laminated mud, suggesting that ice retreated west of the core location, depositing the laminated unit, followed by a readvance that deposited the uppermost diamicton in 146GC. Absence of similar diamictons in 144GC indicates that the margin did not extend seaward of GZW1, and instead implies an oscillating margin likely anchored to GZW2.

## Destabilization of the grounding line and ice-shelf breakup

Between 15.2 and 14 ka BP, a significant shift from a fine-grained laminated mud of LF2 to a gritty, massive sandy silty clay with coarser gravel to pebble-rich layers of Lithofacies 3 (LF3) occurs in both cores. Core 144GC contains two discrete LF3 layers at 98–98.5 and 90–94 cm that consist of a reddish brown coarse sandy-silt containing granules and larger clasts (see Supplementary Fig. 1). The lower boundaries of both layers are sharp. Between these layers is a finer, gritty, massive silt. High magnetic susceptibility measurements (Fig. 3) characterise this section of the core and correspond with the coarse and clast-rich units. From 78 to 90 cm, the stratigraphy is a gritty, sandy clay that fines upwards (Fig. 3). Core 146GC also contains a transition from LF2 to LF3, with alternating layers of fine massive mud and IRD-rich fine massive mud between 350 and 315 cm, prior to 15.2 ka BP. This is then overlain by two thick (34 cm and 200 cm), clast-rich, soft (<10 kPa) diamictons in sandy silty clay. These thicker diamicton layers are faintly stratified and have sharp, convoluted, upper and lower boundaries (Fig. 3).

The foraminiferal assemblage in LF3 (FAZ3) is characterised by continued increase in *C. neoteretis*, reaching a maximum abundance of 50%. *C. reniforme* abundance also increases from 14% at the base of FAZ3 to between 22–25% in the upper section of FAZ3 (Fig. 4). *S. feylingi* and *S. horvathi* decrease from 55–35% to 20–5%. Both the benthic and planktic foraminiferal abundances are generally very high (peaking at 30,000 indiv. cm$^{-2}$ kyr$^{-1}$ for benthic foraminifera and 20,000 indiv. cm$^{-2}$ kyr$^{-1}$ for planktic foraminifera), although abundances are lower within the IRD-rich layers (57 indiv. cm$^{-2}$ kyr$^{-1}$ benthic and 114 indiv. cm$^{-2}$ kyr$^{-1}$ planktic specimens). The $\delta^{18}$O record from planktic foraminifera returns to relatively low values in this unit (generally <2‰). Towards the top of this biostratigraphic unit (88 cm, 12.4 ka BP), *Cibicides lobatulus* increases in abundance to a maximum of 18%.

Overlying LF3 in both cores is a massive brown fine-grained mud (LF4), deposited during the past 10 ka BP (Fig. 3). This lithofacies in core 144GC consists of two foraminiferal zones (FAZ4 and 5). FAZ4 (6.2-10 ka BP) is characterised by a decrease in both *C. neoteretis* and *C. reniforme* and a significant increase in abundance of *S. feylingi* (from 36% to 100%). Overall, the benthic foraminiferal abundance decreases from 4100 indiv. cm$^{-2}$ kyr$^{-1}$ to 108 indiv. cm$^{-2}$ kyr$^{-1}$. Planktic abundances also decrease from 2200 indiv. cm$^{-2}$ kyr$^{-1}$ to 54 indiv. cm$^{-2}$ kyr$^{-1}$ at the top of FAZ4. The planktic $\delta^{18}$O values return to comparatively

high values (3–3.5‰). The overlying foraminiferal assemblage, FAZ5, is characterised by low benthic (<350 indiv. cm$^{-2}$ kyr$^{-1}$) and planktic (<7 indiv. cm$^{-2}$ kyr$^{-1}$) abundances, with the benthic assemblage dominated by agglutinated species with very few calcareous specimens.

The lithostratigraphic and foraminiferal data from cores 144GC and 146GC are interpreted as recording the destabilisation of the grounding line and ice shelf that occurred sometime between 15.2 and 14 ka BP. The coarser units at 94–98 cm and 90–94 cm represent deposition of IRD. A similar lithofacies change is observed in core 135G[22], which occurs at a similar time (14.6 ka BP (13617–15670 yrs BP) age is recalibrated using the method in this paper). This sedimentological and faunal signature is interpreted as evidence of ice shelf breakup, allowing the rafting and deposition of coarse material from icebergs. The transition from LF2 to LF3 in core 144GC is comparable to sub-ice shelf sedimentary sequences reported from Antarctica[35,43], and represents grounding line migration and ice-shelf breakup.

The alternating fine, massive mud and IRD-rich, fine, massive mud layers in 146GC represent a similar sedimentary process to that captured in core 144GC, and are similarly interpreted to mark the retreat of an overlying ice shelf back across the core site[35]. The thick diamicton layers captured in core 146GC are interpreted as debris flow deposits produced during grounding line retreat and redistribution of material upslope of the core site across GZW2. The near synchronous timing of the deposition of LF3 in both cores, which are ~60 km apart, points to a rapid breakup and retreat of the fronting ice shelf and grounding line.

Foraminiferal assemblage data support our interpretation of environmental conditions during this phase of grounding line retreat (Fig. 4). The biostratigraphic change from FAZ2 to FAZ3 indicates a shift from sub-ice shelf conditions to more seasonal sea-ice conditions and a continued strengthening of the RAW influx. This is further supported by the rapid increase in planktic/benthic ratios and planktic concentration, indicating open water and productive surface water conditions[27]. The $\delta^{18}$O record from planktic foraminifera returns to relatively low values (<2‰) through FAZ3, suggesting significant meltwater input from the disintegrating ice shelf and grounding line retreat. The increase in *Cibicides lobatulus* towards the top of FAZ3 (88 cm, 12.4 ka BP) suggests relatively strong currents in a high-energy environment[44] coinciding with the high meltwater signal inferred from the $\delta^{18}$O values. This also coincides with the clast-rich unit (LF3; Fig. 4), with the coarse-grained detritus likely released during ice shelf/sea ice break-up.

The timing of ice-shelf breakup, ~15.2–14 ka BP, corresponds with an increased RAW signal on the continental shelf (increase in *C. neoteretis* shown in 144GC) suggesting that ocean-driven melting played an important role in ice-shelf breakup. Furthermore, the prolonged meltwater signal at this time indicates that there was a large amount of freshwater released from the ice front. This likely acted to enhance the inflow of RAW across the shelf to the grounding line, as has been shown for the present-day 79N[7]. The significant increase in abundance of *C. lobatulus* indicates comparatively strong bottom currents in the area of core 144GC in the latter stages of grounding line retreat. When planktic foraminiferal $\delta^{18}$O values are lowest, *C. neoteretis* and, therefore RAW signal, were high.

After 10 ka BP, the core lithostratigraphy suggests the ice margin retreated away from both core sites, reducing the sedimentation rate and sediment size fraction, with only very fine muds being deposited. The decrease in *C. neoteretis* and *C. reniforme* indicates a reduced RAW influence, whilst the high planktic $\delta^{18}$O values indicate a reduced meltwater influence as the grounding line retreated further from the core site. The increase and dominance of *S. feylingi* through FAZ4 (Fig. 4) indicate unstable conditions with seasonal sea ice present[38]. The overlying foraminiferal record, FAZ5, is characterised by low benthic and planktic concentrations, with only agglutinated species

present in the samples, indicating poor preservation due to dissolution[37]. As foraminifera preservation is so poor in this section of the core, constraining ocean circulation during the mid-to-late Holocene is not possible.

## Discussion

This study provides important insights into the behaviour of the NEGIS following the LGM. We identify internal and external controls on ice-sheet retreat and ice-shelf break up which vary through time.

Evidence from geomorphic features and chronological data suggests that the outer Norske Trough was deglaciated by around 21.6 ka BP[13,18]. This initial deglaciation occurred despite continued cold atmospheric conditions between 20 and 15 ka BP (Fig. 5), when air temperatures were still low (temperature anomaly ~−20 to −15 °C[13,45]). The geometry of the outer shelf, particularly the Norske trough, has been noted as a likely internal control on early deglaciation[22]. The outer Norske Trough forms a retrograde slope, deepening from a relatively shallow shelf edge (320 m BSL) to a maximum depth of 560 m BSL about 100 km from the shelf break (Fig. 1). However, retreat rates from the shelf edge were initially slow, approximately 23 ma$^{-1}$ from the outer shelf to core site 144GC, increasing to 32 m a$^{-1}$ between core site 144GC and DA17/092G[18] (Fig. 1). This is an order of magnitude slower compared to retreat rates estimated from other palaeo-ice streams from the outer shelf of Greenland (e.g., 121 to 94 ma$^{-1}$ for the Ummannaq Ice Stream; 137 to 104 ma$^{-1}$ for Jakobshavn Isbrae[1,13], both in central west Greenland[46]). Water depth alone, therefore, was unlikely to have been the principal control on initial shelf edge retreat. However, Norske Trough narrows from the shelf edge to the mid-shelf (Fig. 1), which would increase lateral drag of the ice stream[47] and slow retreat rates. The relatively slow, episodic retreat of the grounding line is evidenced by GZWs and occasional moraine ridges[22] (Fig. 2).

The more likely dominant driver of initial ice stream retreat in Norske Trough was increased submarine melt of the calving front and sub-ice shelf due to the presence of RAW[48]. Microfaunal evidence suggests that warm Atlantic inflow into the Fram Strait occurred between 19.5 and 14.5 ka BP[49], with evidence of RAW at the continental slope fronting the Norske Trough[19], as well as a strong RAW signal evident at the base of the glacimarine unit in core 144GC (Figs. 4–6). The strong RAW signal in core 144GC also coincides with a significant meltwater input (Fig. 5). Meltwater discharge can regulate the buoyancy-driven circulation that draws warm RAW onto the continental shelf[50,51]. In modern fjord settings, high meltwater input acts to entrain more heat transport to the grounding line, which in turn increases melting at the groundline line[51]. We suggest that such feedback occurred in the Norske Trough during deglaciation. Furthermore, based on our multi-proxy data as well as regional ice-core derived atmospheric temperatures, which indicate cooler conditions between ~20 and 15.5 ka BP, we infer that initial retreat and large meltwater flux were not driven by atmospheric warming but rather by oceanic melting (Figs. 5 and 6). Thus, incursions of RAW to the ice front drove significant melt, increasing buoyancy-driven circulation and strengthening RAW ingress into the sub-ice shelf cavity and to the grounding line.

The geomorphology recorded within the mid-Norske Trough indicates two periods of prolonged stability (GZW1 and GZW2), but the precise timing of their formation is not well constrained by our data. Indeed, it is clear from our acoustic data that an additional, previously unidentified, GZW lies to the immediate east of 144GC (Fig. 6). Prior research has suggested that GZW2 marks the Younger Dryas ice extent[16] or, alternatively, the location of both GZW1 and GZW2 marked stability from around 16.6 ka BP to 13 ka BP[22]. The latter scenario is most consistent with our data, suggesting a prolonged period of stability occurring from approximately 17.6 ka BP until 15.2 ka BP (Fig. 6). Chronological constraint from core 146GC indicates GZW1 was ice-free

before 15.2 ka BP and likely significantly earlier. This period of stability coincided with Greenland Stadial II, when modelled air surface temperatures were still low[45,52] (Fig. 5).

As previously suggested, trough geometry likely played a key role in stabilising the ice margin at GZW2. Firstly, the trough width reduces at GZW2, increasing lateral drag[47] (Fig. 6). Secondly, GZW2 appears to have been formed on a bathymetric high[22] that acted to pin the grounding line. Foraminiferal data from core 144GC indicate a reduced RAW influence (Fig. 5) reaching the grounding line at this time. This study also demonstrates that the NEGIS was fronted by an ice shelf when it stabilised at GZW1 and GZW2. The additional GZW identified to the immediate east of 144GC also indicates the presence of an ice shelf prior to retreat landwards of 144GC (Figs. 1, 2 and 6.) An ice shelf would also promote further grounding line stability by buttressing the ice stream[53]. The maximum extent of the ice shelf is unknown, but if the NEGIS was pinned at the mid-trough GZW2 prior to 15.2 ka BP, then the ice shelf extended at least 60 km to cover both 146GC and 144GC core locations at this time.

The timing of ice-shelf breakup ~15.2–14 ka BP corresponds with an increase in RAW on the continental slope[19] and on the mid to outer shelf (this study, Fig. 5), implying ocean-driven melting was an important driver of breakup (Fig. 6). Initial retreat and ocean-driven melting had a positive feedback effect, increasing the inflow of RAW to the grounding line due to enhanced circulation[51]. However, the timing of the breakup also corresponds with Greenland Interstadial I (Bølling-Allerød). This was a period of rapid increase in surface air temperature[13,45,52] (Fig. 5) and suggests there was also a likely atmospheric influence (via surface melt) on ice-shelf break-up and retreat landward of GZW2. The decrease in sea-ice indicators also supports warmer surface air temperatures. Likewise, Greenland Interstadial I is associated with an increase in the strength of the Atlantic Meridional Overturning Circulation, drawing more warm Atlantic Water northwards into the Fram Strait. Evidence of enhanced Atlantic Water is found in records from both the east and west Fram Strait[17,19,26,54,55] between 14.1 and 12.9 ka BP. It is therefore likely that a combination of increased air temperature and oceanic heat flux from RAW acted in unison with meltwater-RAW feedbacks (buoyancy-driven circulation) to drive retreat of the NEGIS from its mid-trough position. There is no evidence in the foraminiferal data or other data from core 144GC for a Younger Dryas cooling event. Indeed, modelled deglacial ages from core DA17-092G suggest the trough landward of GZW2 was ice-free prior to 12.5 ka BP[26] and therefore GZW2 does not mark a Younger Dryas ice margin.

Core 144GC provides limited evidence of the palaeoceanographic evolution through the Holocene due to poor foraminiferal preservation. However, cores closer to the coastline along the Norske Trough indicate there is a continued presence and influence of RAW on the grounding line as it retreated to the coast. Cores 198GC and 270GC (Fig. 1) both indicate strong inflow of warm recirculated RAW across the continental shelf from as early as 10.9 and 10.1 ka BP, respectively, occurring just after deglaciation of the core sites[24,27]. Evidence of a strong inflow of RAW continues until 7.5 ka BP[24] accompanied by continued grounding-line retreat[13,35]. In combination, the lithostratigraphic and biostratigraphic evidence from cores through the Norske Trough indicates persistent presence and the importance of RAW in driving retreat of the NEGIS from initial deglaciation onwards.

In summary, we have provided further constraints on the timing and drivers of retreat of the palaeo-NEGIS, which extended offshore through the Norske Trough to the continental shelf edge at the LGM. Whilst surface air temperature undoubtedly played a key role in forcing retreat during the latter stages of deglaciation, the flow of RAW to the margins of the NEGIS was the primary control on grounding line stability and ice shelf presence/absence. Meltwater-related feedbacks and boundary conditions, such as bathymetry (i.e.,

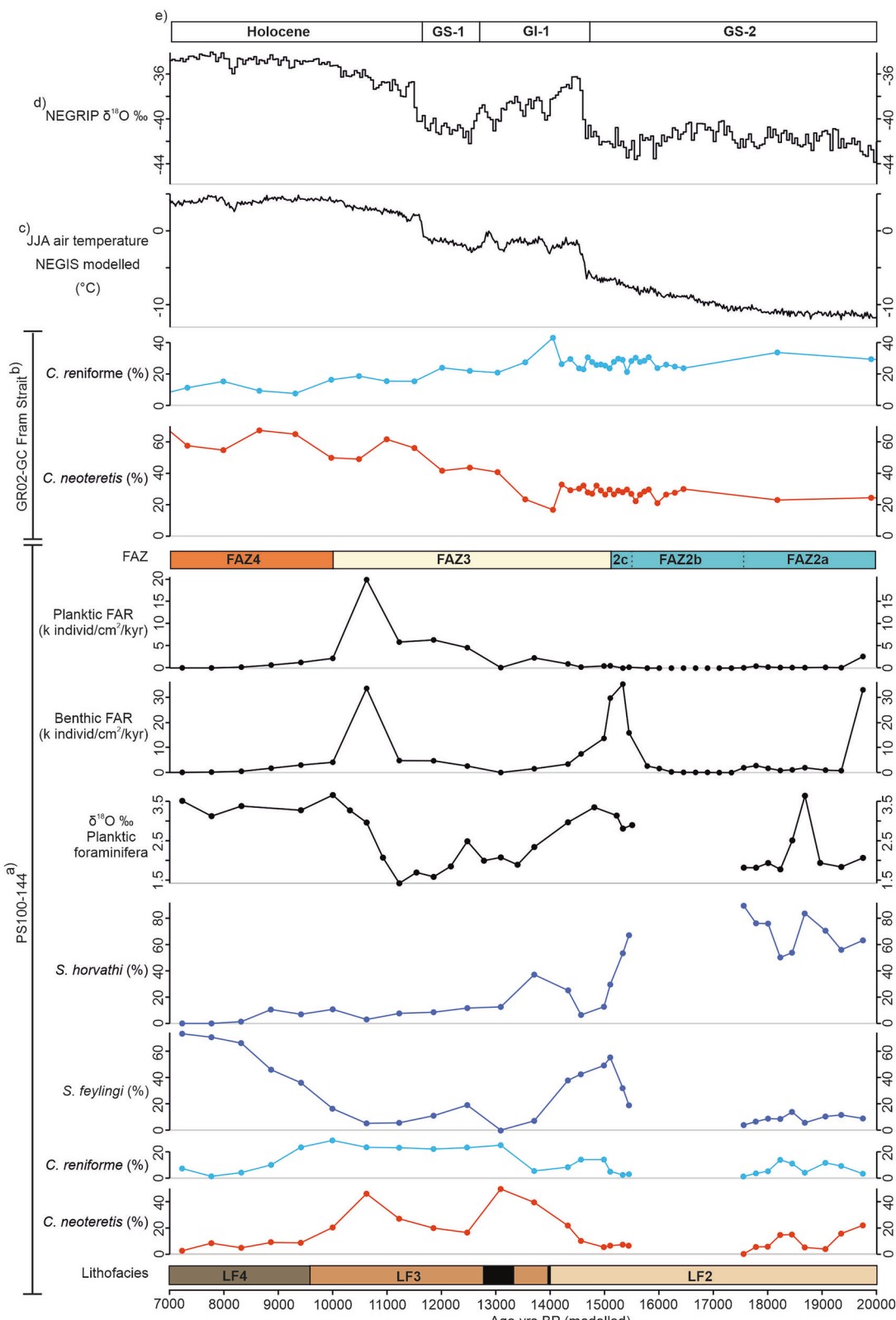

**Fig. 5 | Comparative plots of key proxies from this study with previous published proxy and modelled climate record from northeast Greenland.**
**a** Summary of benthic foraminiferal data, as well as planktic δ18O record and planktic and benthic foraminiferal accumulation rate (FAR) for core 144GC. The lithofacies zones (LF2-LF4) are included with the Ice Rafted Debris (IRD) rich layers identified in the core stratigraphy represented by the thick black lines. The foraminiferal assemblage zones FAZ2-4 are also included. Note the uppermost section of the core record is not included here due to poor chronological control and

foraminifera preservation; **b** summary of foraminiferal data from core GR02-GC in the Fram Strait[19]; **c** modelled summer (June, July, August; JJA) air temperature at Zachariae Isstrom, 78.9°N, 20.5°W, 103 masl[52]; **d** NEGRIP δ18O ‰[60]; **e** showing the timing of the climate periods constrained by the Greenland ice core record and include Greenland Stadial 2 (GS-2, period including the LGM and Heinrich Stadial 1), Greenland Interstadial 1 (GI-1, Bølling-Allerød), Greenland Stadial 1 (GS-1, Younger Dryas), and the Holocene.

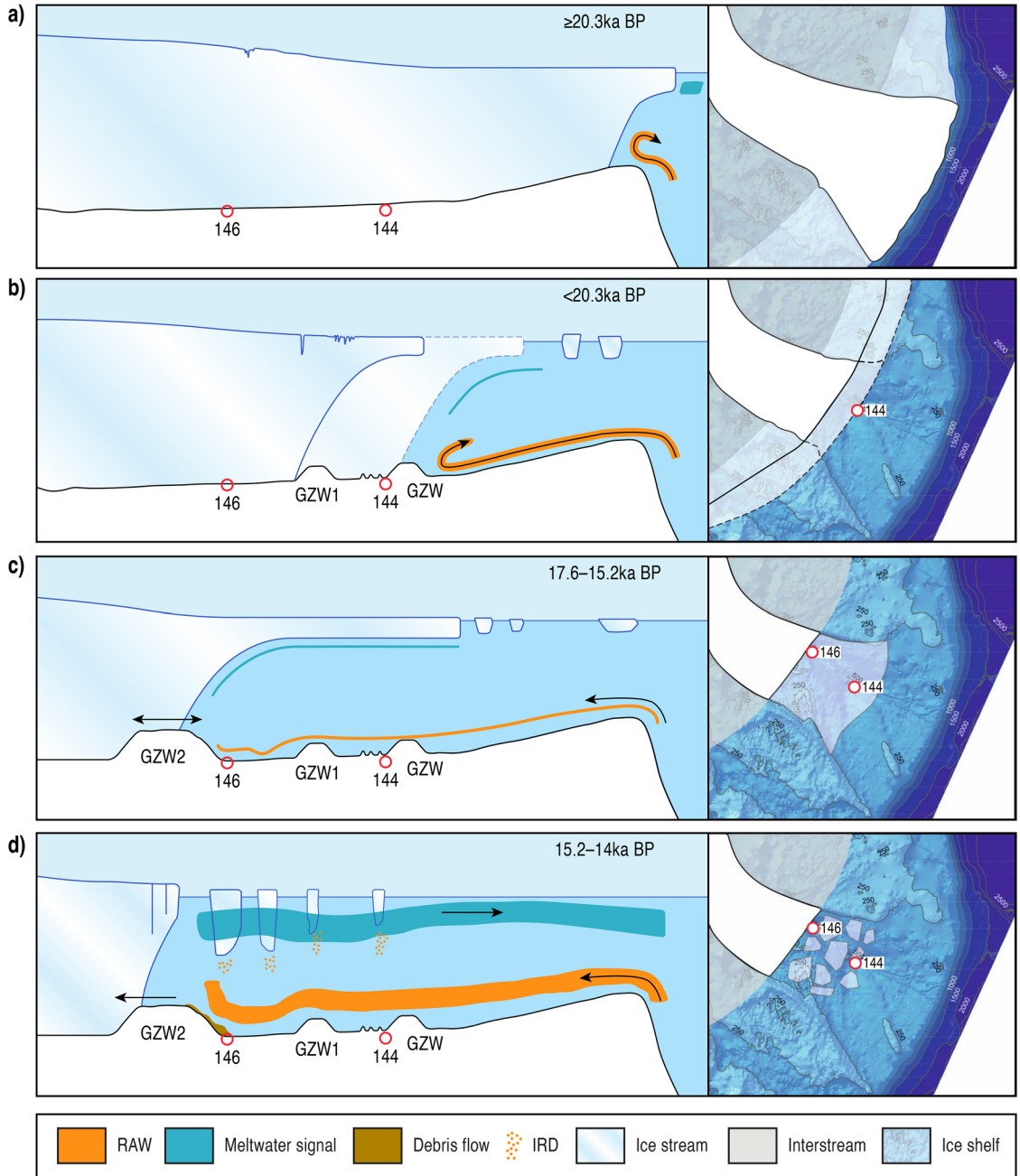

**Fig. 6 | Conceptual model showing the Northeast Greenland Ice Stream (NEGIS) ice-ocean interactions through time.** The conceptual model represents the interaction between the marginal position of the NEGIS, ocean circulation (Return Atlantic Current (RAW)), ice shelf and Ice Rafted Debris (IRD) activity. **a**–**d** represent key time periods discussed in the manuscript and consist of a cross-sectional view of the NEGIS and the fronting ice shelf, as well as a top-down view showing extent in relation to the current NEGIS continental shelf. Bathymetry, position of Grounding Zone Wedges (GZWs) and position of sediment cores analysed are illustrated and are estimates. The ocean bathymetry is from GEBCO[57].

trough narrowing), also played a part in both amplifying ocean-melting and slowing down retreat. This work emphasises the need for detailed ocean observations near the contemporary grounding line, as well as detailed knowledge of along-flow bathymetry, in order to predict the future behaviour of this and other marine-based ice streams.

## Methods
### Marine geophysical analysis
Sub-bottom profiler data was collected using a hull-mounted Para-sound DS III-P70 system operating at a pulse mode of between 4 and 20 kHz and a pulse length of 0.5 ms. The seismic profiles were visualised using PS3 file formats on SeNT v 2.02. This data was used to guide core site locations.

### Core collection and CTD
Marine geophysical data and gravity cores were collected across the northeast Greenland continental shelf during cruise PS100 on the *RV Polarstern* in 2016. The cores were recovered from the outer section of Norske Trough within a basin that lies between the shelf edge and the GZW2. Core PS100-144GC (77 7.495N, 10 34.239W, 496 m bsl) lies within a basin located between the outer shelf edge and GZW1 (Fig. 1b). The total length of this core is 216 cm. Core PS100-146GC (77 31.381N, 12 20.131W, 505 m bsl) is

**Table 1 | Radiocarbon dates from sediment cores discussed in the text**

| Lab code | Depth (cm) | Material | $^{14}$C age ± 1σ (yrs BP) | Calibrated Age using (cal yrs BP)(2σ upper and lower range) | $\Delta R^{GS}_{2\sigma}$ | Calibrated Age GS (cal yrs BP) (2σ upper and lower range) | Calibrated age combined range and median |
|---|---|---|---|---|---|---|---|
| **PS100-144** | | | | | | | |
| UCIAMS-210585 | 72–73 | Mixed benthic foraminifera | 9370 ±70 | 10,003 (9663–10,271) | | | |
| SUERC-76494 | 104–105 | Mixed benthic foraminifera | 13,622 ± 44 | 15,586 (15,284–15,887) | 832 ± 66 | 14,361 (14,046–14,771) | 14,966 (14,046–15,887) |
| UCIAMS-210586 | 174–175 | Mixed benthic foraminifera | 16,520 ± 120 | 19,042 (18705–19420) | 832 ± 66 | 18,184 (17,800–18,572) | 18,610 (17,800–19,420) |
| **PS100-146** | | | | | | | |
| UCIAMS-216441 | 72–73 | Mixed benthic foraminifera | 10,005 ± 30 | 10,896 (10,698–11,086) | | | |
| SUERC-79058 | 288–289 | Mixed benthic foraminifera | 13,453 ± 46 | 15,365 (15,081–15,656) | 832 ± 66 | 14076 (13,779–14,418) | 14,718 (13,779–15,656) |
| UCIAMS-211064 | 333–334 | Mixed benthic foraminifera | 13,790 ± 50 | 15,810 (15,508–16,114) | 832 ± 66 | 14,653 (14,265–15,001) | 15190 (14,265–16,114) |

Estimation of marine reservoir correction follows the approach outlined for polar regions in Heaton et al. (2023). The local reservoir correction (ΔR) and error (ΔR$_{error}$) for marine reservoir correction used in calibrating Holocene-aged samples is estimated based on average of 8 nearest points from the radiocarbon reservoir correction database (Reimer and Reimer, 2001) giving a value for the local Holocene reservoir correction (ΔR$^{Hol}_{2\sigma}$) of 2 ± 66. For pre-Holocene samples the maximal $^{14}$C glacial depletion is taken from Heaton et al. (2023) and added to ΔR$^{Hol}_{2\sigma}$ to produce the glacial correction (ΔR$^{GS}_{2\sigma}$). We then use the combined range and median age of that range in the discussion.

located on the outer/seaward edge of GZW2 and is 575 cm long. Each core was cut into 1 m sections, split, photographed, and described onboard. Core description recorded information on grain size, colour, sorting, bed contacts, clast abundance, sedimentary structures, and the presence of macrofossils. Shear strength measurements, using a handheld Torvane, were also taken at 10 cm intervals down core.

### Physical properties analysis
Post-cruise physical property analysis was conducted on all cores at Durham University. Each core was X-rayed using a GEOTEK CT scanner. The X-rays were used to augment the core descriptions for more detailed lithofacies analysis and allowed clearer identification of IRD content. Magnetic susceptibility, wet bulk density and p-wave velocity measurements were collected using a GEOTEK Multi-Sensor Core Logger at 1 cm resolution down core. Additionally, 47 samples were collected from core PS100-144GC for particle size analysis using a laser particle size analyzer. Samples were prepared by removing organic material using 10% Hydrogen peroxide. Sodium hexametaphosphate was used to defloculate the sediment, and samples were sieved through a 2 mm mesh sieve prior to putting through the laser scanner. This data was used to calculate the proportion of clay, silt, and sand-sized particles.

### Foraminiferal analysis
Detailed foraminiferal analysis was conducted on core PS100-144GC with 46 samples analysed down core. Samples were volumetrically measured, then wet-sieved through a 500 μm and 63 μm sieve to remove the coarse and fine material. The 63–500 μm fraction was used for foraminiferal analysis. Each sample was counted wet and on the same day as sample preparation to prevent the dissolution of small foraminiferal species and destruction of agglutinated species. Absolute abundances of both benthic and planktic foraminiferal counts are presented as both concentrations (number of species per 1cc of sediment; Fig. 4) and as foraminiferal accumulation rates (FAR) or fluxes. FAR (as individuals cm$^{-2}$ ky$^{-1}$) was calculated using the following equation:

Equation 1: FAR = TSAR∗FN

Where TSAR is the total sediment accumulation rate (g cm$^{-2}$ kyr$^{-1}$) and FN the number of foraminifera per gram[56].

### Foraminiferal stable isotope analysis
Oxygen and carbon isotope analysis was conducted on the benthic foraminifera *Cassidulina neoteritis* (28 samples) and planktic *Neogloboquadrina pachyderma* (32 samples). Analysis was conducted at the NERC Isotope Geosciences Laboratory at the British Geological Survey. Analysis was conducted using an IsoPrime mass spectrometer with a Multicarb preparation system. Stable isotope results were calibrated to the VPDB scale of international standards with an analytical precision of >±0.05‰.

### Chronology
The chronology of the two cores is based on 6 accelerator mass spectrometry (AMS) $^{14}$C ages (Table 1) measured on mixed species of benthic foraminifera although Miliolina and Quinqueloculina species were avoided. AMS $^{14}$C samples were submitted to the NERC radiocarbon facility in East Kilbride, where they were prepared to graphite and passed to the SUERC AMS laboratory (SUERC publication codes) or the Keck C Cycle AMS laboratory, University of California, Irvine (UCIAMS publication codes) for $^{14}$C measurement. The radiocarbon dates were corrected for isotopic fractionation and then calibrated to calendar years (cal a BP) using Calib 8.20 (Stuiver and Reimer, 1993) and the Marine20 radiocarbon calibration curve (Heaton et al., 2020). We have followed the approach presented in Heaton et al. (2023) for calibrating in polar regions to calibrate the ages presented here. This

takes into account the temporal uncertainties, with most likely larger reservoir ages in the marine reservoir correction for high latitudes during glacial periods. For radiocarbon dates with a calibrated age younger than the start of the Holocene (taken here as from 11,500 cal a BP), we calculate $\Delta R$ and $\Delta R_{error}$ based on the 8 closest samples from the Marine Reservoir Correction Database (http://calib.or g/marine/, Reimer and Reimer, 2001). This gives a marine reservoir correction ($\Delta R^{Hol}_{20}$) of 2 ± 66 used in calibration (Table 1). For pre-Holocene-aged samples, we perform two calibrations to provide a more realistic estimate of the age and age uncertainty. The first calibration assumes minimal polar $^{14}C$ glacial depletion and uses the $\Delta R^{Hol}_{20}$ estimate as above. The second calibration uses an estimated latitude-dependent maximal polar $^{14}C$ glacial depletion from Heaton et al. (2023). This is added to the Holocene value to give the glacial correction: $\Delta R^{GS}_{20}$. We then take the conservative approach of using the full calibrated range from these two calibrations to calculate the median and range of ages used in the discussion (all calibrations are shown in Table 1).

The three calibrated AMS ages from core PS100-144GC were then used in the Bayesian accumulation age-depth modelling program *Bacon 2.2* to develop an age-depth model. We performed two age-depth models. The first using the $\Delta R^{Hol}_{20}$ to calibrate all ages, the second using $\Delta R^{GS}_{20}$ the for the two older ages as described above (Supplementary Figs. 3 and 4). We then took the average of the two models for each cm to develop the age-depth model used in this study. The lowest radiocarbon date from PS100-144GC constrains the base of the glacimarine unit at 190 cm to 20347 cal yr BP (full range of 18673–22689). Due to the poor chronological control in the upper section of the core, an assumed age of 0 was used for the core top. This is likely to be incorrect due to loss of the surface sediments through the coring process. We therefore only compare the proxy datasets from 20,000-10,000 cal yr BP, where the age model is more confidently constrained. No age-depth model was produced for core 146GC due to the limited chronological control constraining the older section of the core. Foraminiferal abundance was too low to allow additional radiocarbon ages to be measured.

## Data availability

The datasets used in this study are available at the UK NERC Polar Data Centre (http://www.bas.ac.uk/data/uk-pdc/).

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

## Acknowledgements

This work was funded by the UK Natural Environment Research Council (NERC) grant NE/N011228/1 to D.H.R., C.Ó.C., J.M.L and J.A.S. The core chronology was supported by the NERC radiocarbon grant 2113.0418. D.H.R., C.Ó.C., J.M.L and S.L.C. We thank the Officers and crews of the RV Polarstern during cruise PS100 for their excellent support. We gratefully acknowledge support from the Alfred Wegener Institute (AWI) for ship time via grant AWI_PS100_01 to D.H.R. We thank Chris Orton for the assistance with Fig. 6. We thank Neil Tunstall and Chris Longley for assistance with Geotek core scanning at Durham University.

## Author contributions

S.L.C.: Principal writer, writing, review and editing, Conceptualisation, Investigation (Cruise participation), Data analysis and interpretation, Figure preparation. D.H.R.: NEGIS Project PI, Conceptualisation, Funding acquisition, Investigation (Cruise Participation), writing, review and editing, Data analysis and interpretation; C.Ó.C. and J.M.L.: Writing, review & editing, Investigation (Cruise Participation), Data analysis and interpretation, Conceptualisation, Funding acquisition; J.A.S.: Writing review and interpretation, Funding acquisition; C.A.G.: Review, Data processing and figure preparation; T.K.: Review and editing, Investigation (cruise participation).

## Competing interests

The authors declare no competing interests.
