## [Transparent Peer Review file · Nature Communications]

Ocean driven retreat of the Northeast Greenland Ice Stream following the Last Glacial Maximum.

Corresponding Author: Dr S. Louise Callard

Version 0:

Reviewer comments:

Reviewer #1

(Remarks to the Author)

This manuscript presents new data on the history of the NEGIS from the LGM through early Holocene. It builds on previous work that has shown that the ice margin reached the shelf edge at the Westwind Trough and that it had already retreated off the outer shelf during the LGM. What this manuscript adds is faunal evidence from a key core (144GC) and lithofacies analysis plus dating on two cores (144 and 146GC) to gain more detailed understanding of the further retreat history and what forcings were at play. The core sites target important ice marginal features (grounding zone wedges) on the mid/outer shelf. The lithofacies and foraminifera assemblage data are used to reconstruct important aspects of the NEGIS in the past: the configuration of the ice front (i.e. ice shelf fronted or ice cliff fronted); emissions of meltwater, advection of relatively warm RAC water, and sea-ice cover. I was struck by two key findings. The first is firm indication that the ice retreated during the LGM and during the oldest part of deglaciation, HS-1. The cores do not contain a complete record of this retreat as they were ice covered until c. 19 ka. A basal diamicton interpreted as subglacial deposit is overlain by fine grained laminated sediment lacking ice rafted clasts and containing a fauna consistent with an ice shelf fronted margin. The second key result to my mind is that the GZWs record relative stability of the grounding zone during the interval of c. 19ka to c. 15.8 ka. After c. 15.2 ka, the lithofacies indicate a calving margin lacking an ice shelf and the faunas indicate strong inflow of RAC and relative ocean warmth at the seabed that would influence ice instability. The GZWs had been thought to be Younger Dryas in age, so this is a key finding! The conclusion is that ice retreat from the GZW 2 may have been driven both by warm atmosphere and ocean forcing. I think that the data exist in this paper to make these points. However, there are some structural problems and some lack of clarity in the manuscript that hurt its ability to make these points clearly.

The Results section does not clearly distinguish previous work and background information from new results presented in this study. I would argue that the first 3 paragraphs are actually part of an introduction, setting the stage for the problems that the core data can attempt to solve. The fourth paragraph is meant to indicate how the lithofacies and foraminifera in the two cores can be associated with environments, but it was confusing because it goes immediately to interpretations without presenting the characteristics of the lithofacies and fauna and how these progress upcore and in association with the chronology. In a sense, there is no results section in this paper. The data are not described in a systematic way before interpretations are made. It is difficult to separate what is new and what is previously published.

Line 189 there is an error in the sedimentation rate calculation...I think it is off by two orders of magnitude. Maybe what is meant is 0.2 cm per year rather than per kyr.

The figures are nicely drawn, but they fall short of helping to convey the points being made in some cases.

On Figure 1, the map of the area, two GZWs are shown. Core 144 is shown immediately in front of the outer GZW. I was however, confused that on the acoustic profile of the core site in Figure 3, that core 144 is shown to the NW of GZW1. Is this a matter of the directions being misplaced on the figure? I am unsure, but think this profile needs to be turned around. It would help as well if Figure 1 showed where the two profiles are in the trough so that the orientation and locations are clear. In addition, it would help the introduction if there was a panel added to show the modern NEGIS, maybe using flow speed. Otherwise it is not easy for uninitiated readers to match the outlet glaciers with the main ice stream.

Figure 4 could benefit from having the Lithofacies along the depth axis as in Figure 2 so a reader does not need to flip back and forth. Suggest you include planktic forams per ml and benthic forams per ml. You could use the planks per ml instead of the P:B ratio if space is an issue. The ice species are in purple not blue...at least in the reviewer pdf. I presume that faunal analyses were not made on core 146 which is too bad.

Figure 5. needs some additional annotation such as chrons (LGM, HS1, B-A, YD etc). Maybe add the modern average air temperature to the summer air temp panel. Might be useful as well to tie in the faunal zones and lithofacies as these are

important as well as the fauna.

Figure 6 shows schematic profiles of the reconstructed NEGIS through time. Panel a indicates that the ice margin retreated by calving back and that dirty ice bergs were released. Early in the paper there is evidence of IRD in slope cores during this period. Panel b is the starting point of this study. But this panel does not match the data. GZW 1 is a bump on the seafloor but no association made with the core. Is GZW 1 associated with deposition of the laminated facies lacking IRD? Does that suggest that this is an ice shelf fronted margin beginning from the very start of the core record? This schematic does not explain GZW 1 and how it relates to results presented for 144GC. Panel C makes sense except that the story either is missing the IRD interval of ice retreat to GZW 1, or that the formation of GZW 1 is ignored. Presumably the GZWs are formed as the ice stabilizes, so perhaps they should not appear until the panel in which they form.

Reviewer #2

(Remarks to the Author)

Review of Callard et al (2025) - 'Ocean driven retreat of the Northeast Greenland Ice Stream following the Last Glacial Maximum'

Summary

This study presents data and conclusions from two new sediment cores taken on the Norske Trough that are used to describe the early deglaciation of the Northeast Greenland Ice Stream (NEGIS) from its LGM position. The paper has numerous interesting findings that are used to describe the evolution of the paleo-ice stream by tying in sediment facies, analysis of foraminifera assemblages and glacial landforms. Chief among them is the important role that ocean melt, driven by warm Return Atlantic Water Current (RAC), played in the initial retreat of NEGIS from its LGM by ~20 ka BP, a retreat that occurred without a rise in atmospheric temperature and that was much earlier than previously thought. RAC is further linked with the collapse of the ice shelf after 15.7 ka BP when surface air temperatures began to rise during the onset of the Bølling-Allerød. Furthermore, Grounding Zone Wedges in the Norske Trough provided vital pinning points and stability during the early deglaciation after the trough deepens into a retrograde slope inland of the shelf edge.

I found this paper very enjoyable to read and believe it shares valuable insights into the evolution of the NEGIS from the LGM. This is especially topical as constraining the future response of the contemporary NEGIS to climate warming is crucial. The manuscript is well-written and well-structured, with the authors clearly explaining their interpretations of sediment record which I, for the most part, agree with.

My comments below mainly relate to interpretations made around the early retreat/deglaciation of the NEGIS, specifically whether an ice shelf existed before 18 ka BP, and the role the bathymetry and fjord topographically played in the initial retreat. I will state here that my opinions are far from set in stone, and hopeful adding a few sentences to further explain their interpretations would alleviate any of my concerns and lead me to recommend this paper for publication.

Major Comments

- Ice Shelf Interpretation: I agree with the authors interpretation that an ice-shelf existed in the Norske Trough with the grounding line on GZW2 between 18-15ka BP. However, I'm curious why the authors believe an ice shelf wasn't present before NEGIS retreat to GZW2 (for example in Figure 6B). Instead, during the early deglaciation (>20 ka BP to 18 ka BP), the high levels of *S. horvathii* and lack of IRD are presented as reasons for thick sea-ice conditions. These same arguments are then used as evidence for an ice shelf between 18-15 ka BP. Furthermore, as the authors rightly mention, the occurrence of GZWs are indications of shelf ice conditions, therefore I would expect it likely that an ice shelf was present with the grounding line on GZW1 during the early deglaciation (López-Quirós et al., 2024). I also wonder why the keel marks from tabular ice bergs must have been formed after 18 ka BP, could they not have been formed from an earlier ice shelf?

Later in the discussion it is stated '... that NEGIS was fronted by an ice shelf when it stabilised on GZW2'. This is not what is shown in Figure 6B when the retreat is shown as a calving cliff face with harsh sea-ice conditions when it reached GZW2. It is then said that the presence of an ice shelf was of 'negligible' importance to the initial stability of NEGIS and that the bathymetric high was more important. I think that a reference or two is required here, as I'm not entirely convinced by that conclusion.

- Bathymetry and Fjord Geometry: I agree with the authors that RAC was likely the cause of early retreat and echo that this is a great finding of the paper. However, I'm not sure I fully agree with the authors comments on the fjord geometry and slope angle regarding this initial retreat, where the initial slow retreat rates (23 to 19 m/yr) on a retrograde slope are used to conclude the bathymetry was not a principal control. Firstly, I think it would be great to have a comparison against retreat rates from other paleo-ice streams in Greenland, is 23 m/yr considerably slower? Secondly, I see that the Norske Trough immediately narrows from the shelf edge, could this have slowed initial retreat through increased lateral drag, similarly as the authors later discuss with the narrowing of the trough around GZW2. Finally, while the bathymetry deepens from 320 m at the shelf edge to 560 m 100 km in land, the majority of this deepening (210 m) happens ~50km from the shelf edge, therefore, wouldn't it be reasonable to while retrograde in nature, the initial shallow slope may have contributed to the slower retreat rates?

Minor Comments

- Figure 1: I think the polygons for GZW1 and 2 should only have the outlines and not be filled, it would then be possible to see how the GZWs look in comparison to the sea floor. I also wonder if the scale for the bathymetry could be clipped to range from 0 to 1500 m, just to make the variations in depth slightly clearer.

- Figure 5: JJA air temperatures. I think this should be made clear is if for the contemporary NEGIS region or the paleo domain of the ice stream? I'm also curious if the temperatures are any different in the newer Badgeley et al., 2020 dataset?
- l.34: It would be worthwhile mentioning that 79N still has an ice tongue that is providing significant buttressing to the interior ice (Humbert et al., 2023), especially as you discuss ice shelf evolution and collapse in the results/discussion.
- l.24: ice shelf break 'up'?
- l.29: Redefine the Greenland Ice Sheet acronym - GrIS.
- l.36: Capital W for 'with'.
- l.40 – 42: Sentence is a bit sloppy. 'There' should be 'their', and maybe reads better as '... their reliability depends on an accurate parametrisation of ice-sheet physics, including key boundary conditions such as...'
- l.45: 'enabling a better understanding of ice-ocean interactions' – We already have some understanding of how ice-ocean, but the paleo record definitely improves this.
- l.51: double 'from' – maybe better as 'Recent evidence from marine cores taken on the continental shelf...'
- l.57: '... initial retreat of NEGIS and critically...'
- l.94: I am presuming that the moraine ridge referred to here as not GZW1? I think this could be made a little clearer by also stating that 144GC is also in front of GZW1.
- L.138: Missing full stop.
- l.146 where is core CR02-GC? I don't see it in Figure 1.
- l.149: Incorrect capitalisation of D.
- l.164: Full stop within the brackets when it should be outside.
- l.180: Add reference to Fig.4 at the end of the sentence.
- l.258: Missing bracket
- l.259: 'Driven melting' and 'driven ice shelf breakup', choice of words could be improved in this sentence.
- l.282: Missing full stop.
- l.287: Missing full stop.
- L.291: 23 m/yr is for the Norske Trough while 19 m/yr is the Westwind trough in reference 20. This should be made clearer (Ó Cofaigh et al., 2025).
- l.334: 'melting was an important driver'
- l.353: Should AWI not be AW?
- l.353: Missing spaces and doubles spaces in this sentence.

References

- Badgeley, J.A., Steig, E.J., Hakim, G.J., Fudge, T.J., 2020. Greenland temperature and precipitation over the last 20000 years using data assimilation. *Clim. Past* 16, 1325–1346. <https://doi.org/10.5194/cp-16-1325-2020>
- Humbert, A., Helm, V., Neckel, N., Zeising, O., Rückamp, M., Khan, S.A., Loebel, E., Brauchle, J., Stebner, K., Gross, D., Sondershaus, R., Müller, R., 2023. Precursor of disintegration of Greenland's largest floating ice tongue. *The Cryosphere* 17, 2851–2870. <https://doi.org/10.5194/tc-17-2851-2023>
- López-Quirós, A., Junna, T., Davies, J., Andresen, K.J., Nielsen, T., Haghpor, N., Wacker, L., Olsen Alstrup, A.K., Munk, O.L., Rasmussen, T.L., Pearce, C., Seidenkrantz, M.-S., 2024. Retreat patterns and dynamics of the former Norske Trough ice stream (NE Greenland): An integrated geomorphological and sedimentological approach. *Quat. Sci. Rev.* 325, 108477. <https://doi.org/10.1016/j.quascirev.2023.108477>
- Ó Cofaigh, C., Lloyd, J.M., Callard, S.L., Gebhardt, C., Streuff, K.T., Dorschel, B., Smith, J.A., Lane, T.P., Jamieson, S.S.R., Kanzow, T., Roberts, D.H., 2025. Shelf-edge glaciation offshore of northeast Greenland during the last glacial maximum and timing of initial ice-sheet retreat. *Quat. Sci. Rev.* 359, 109326. <https://doi.org/10.1016/j.quascirev.2025.109326>

Jamie Barnett
Stockholm University

Reviewer #3

(Remarks to the Author)

Callard et al., provide new information relating to the role oceans played in governing the retreat of the Northeast Greenland Ice Stream on the mid/outer continental shelf. Key results are:

- Deglaciation began at around 20ka BP on the mid/outer Northeast continental shelf, this is earlier than previously reported in the Norske Trough.
- Analysis of foraminiferal assemblages suggests this was primarily driven by the intrusion of warm Return Atlantic Water onto the shelf, amplified by a retrograde seabed.

Overall, I liked that paper, and I found it interesting; the discussion was enjoyable to read. I believe that the contribution is important in this region, particularly given the impact of warming ocean waters on the fate of the Greenland Ice Sheet today and in the future. I think that the methodologies and science are sound, I have some comments on this in the 'main comments' section of this review.

One of the main comments I have relates to the quality of the writing and mistakes throughout. I believe that these detract from the overall manuscript. I therefore list those small changes first here. However, I recommend that the entire manuscript be carefully proofread edited prior to subsequent review.

Comments relating to quality of writing and figures

- Sometimes the authors use 'the' NEGIS sometimes 'the' is excluded – it should be consistent

- Type in figure 1a caption – ‘value’ coldest, I think this should be blue.
- Line 41 – their reliability not there reliability
- RAC never defined. I think it would be more correct to use Return Atlantic Water (RAW) to describe the water mass in some instances. E.g. like 309. This should be checked throughout
- The full core name of 135G should be given in the figure caption of 1b, as per the other cores listed
- What is AWI Bank – not defined anywhere
- The Rasmussen et al., 2022 core should be marked on the map (19G), it is in the caption but not on the map. 39G is also listed in the caption but not on the map. Please check all of these again.
- I think fig 2 and 3 should be switched as fig 3 is introduced in the text first and this is confusing
- Fig 4: foraminifera species names should always be in italics, also in Fig 5
- Fig 4 caption, should this not be <300 counts
- Please check the spelling of all species throughout the manuscript. I noticed some mistakes (e.g. s. hovathi has two l's on line 141)
- No references linked to *S. feylingi* inferences – it is my understanding that this species is linked to productivity (e.g. Seidenkrantz, 2013).
- Typo line 149 – capital D
- Space missing line 258 after 144GC
- Typo line 259, remove driven
- Full stop missing from line 282
- Full stop missing line 287
- I would start a new paragraph for “as previously.” (line 320).
- Foraminifera should be foraminiferal when it is used as an adjective (e.g. foraminiferal assemblages, line 323 and foraminiferal species, line 409). This should be checked throughout the manuscript and changed accordingly
- References required for ice shelf statement – line 328
- What is AWI (Line 353)
- Indicate not indicates (line 353)
- Space after 270GC (Line 254)
- Indicate not indicated (Line 358)
- Constraints not constrains (Line 360)
- Extra l before planktic in fig caption 5a
- I think it would be better to be consistent using line graphs or bar charts for the foraminifera data in Fig 4 and 5
- The end of Figure 6 caption needs removing. And the correct description of the figure included.

Main comments

- I very much enjoyed reading the discussion, I think the argument for RAC is strong with the nuances discussed and previous evidence well incorporated!
- There is a wealth of recent literature highlighting past changes to the NEGIS, for me, the fact that deglaciation began earlier than previously reported needs to be stated in the abstract as I think it is a very important finding and differentiates this paper from those previously published. This is clear in your summary at the end of the paper (Line 360 start), but could be better stated in the abstract. Simply adding ‘Radiocarbon dating shows deglaciation began ~20 ka BP, earlier than previously reported, etc’ would suffice.
- I think it is important to present the foraminiferal data as concentrations, (e.g. no. taxa/g dry sed), given the impact of compaction downcore. As volumetric data associated with the samples (as listed in the methods) was taken, I think it think this is possible if bulk sediment weights are/were made. If this is not possible, I think foraminifera fluxes could be useful to include.
- I think that this paper should be cited, given that it examines the distribution of modern benthic foraminifera in the exact study area being investigated here and the proxies used:
 - o Davies, J., Lloyd, J., Pearce, C. and Seidenkrantz, M.S., 2023. Distribution of modern benthic foraminiferal assemblages across the Northeast Greenland continental shelf. *Marine Micropaleontology*, 184, p.102273. Useful inferences about specific species should be added from this (e.g. *Stetsonia horvathi*)
- Fig. 2. I do not understand the reasoning behind the order of the xray images in the red box. I think it make more sense to list this in order of appearance in the cores. It would also be helpful if the lithofacies abbreviations were defined in the caption.
- I think this reference is also important to include: Tabone, I., Robinson, A., Montoya, M. and Alvarez-Solas, J., 2024. Holocene thinning in central Greenland controlled by the Northeast Greenland Ice Stream. *Nature communications*, 15(1), p.6434. It provides useful information relating to the importance of constraining timing and speed of retreat of the NEGIS and I believe would be useful for the introduction.
- To me it seems like these cores are located on the mid/outer shelf, I wouldn't describe 146GC as being on the outer continental shelf (Line 71).
- Given the high abundance of *C. reniforme*, I think more focus needs to be given to its interpretation in the results and discussion. Broadly, it seems to covary with *C. neoteretis* and previous studies have linked it to chilled Atlantic water. I think this should be included and discussed.
- I would find it helpful to have the Faunal zones marked on Fig 5.
- Why is the age model for 146GC not included (e.g. as part of supplementary figure 3)?
- Were all benthic foraminifera species included in the radiocarbon samples? Or were some species (e.g. *Miliolina*) excluded. This is important to outline these details in the methodology.

Version 1:

Reviewer comments:

Reviewer #1

(Remarks to the Author)

The revised manuscript is much better and I do not have any additional comments or suggestions. I think it is well written and illustrated and the conclusions are noteworthy.

Reviewer #2

(Remarks to the Author)

2nd Review of Callard et al (2025) - 'Ocean driven retreat of the Northeast Greenland Ice Stream following the Last Glacial Maximum'

Summary:

The authors have done an excellent job in responding to the comments from myself and the other reviews. This new version of the manuscript is both an improvement in structure and writing; while also further emphasising, valuable insights gained from the paleo-evolution of NEGIS and crucially the importance of ocean melt in the stability of ice shelves and grounding line behaviour.

My previous comments mainly concerned the interpretations of glacial dynamics at the beginning of the deglaciation, whether an ice shelf existed during retreat from the shelf edge, as well as the role of bathymetry in driving retreat. Both concerns have been alleviated through added references and greater context in the discussion, as well as improvements to Figures 1 and 6. Furthermore, there are numerous small changes throughout the manuscript that address all my minor comments completely. I would also like highlight the restructuring of the manuscript to include the 'Background to study site' section after the Introduction. This was in response to Reviewer #1, and I believe this greatly improves the readability of the manuscript and enhances the story of NEGIS retreat of the LGM and the importance of warm Atlantic water.

To conclude, I believe this piece of work provides novel and crucial insight into the deglaciation of the Greenland Ice Sheet from the LGM and importantly the role of warm Atlantic water is driving grounding line retreat and ice shelf instability. The early retreat of the NEGIS (~20 ka BP) despite a lack of atmospheric warming, plus the enhanced Atlantic water signal around the collapse of the ice shelf are vital conclusions that stress the importance of the ice-ocean boundary as we strive to understand the retreat of contemporary ice sheets through both observations and modelling efforts.

I just have a few minor comments relating to grammar:

- In the response to reviewers there is mention of a "Background to Study Site" section, however I can't see this sub-heading. Maybe an issue with the PDF but I want to highlight in case.
- Line 31-32: Maybe could be re-written to avoid saying the Greenland Ice Sheet twice in quick succession.
- Line 195: Missing space between cores and 114GC
- Line 217: Comma should be a full stop
- Line 349: Missing space after 'approximately'
- Line 357: Missing space after GZW
- Line 357: Miss placed bracket after "moraine ridges"
- Line 364: Missing bracket and full stop after Fig 5.
- Line 393: Missing bracket before Figs

Jamie Barnett

Stockholm University

Reviewer #3

(Remarks to the Author)

I think the author's have adequately addressed the comments made on their manuscript. However, I could not see this reference in the reference list:

References required for ice shelf statement – line 328

We have included reference 53: Gudmundsson, G. Ice-shelf buttressing and the stability of marine ice sheets. *The Cryosphere* 7, 647-655 (2013).

I could not see this reference in the text. Reference 53 refers to Aagaard-Sørensen, S. et al. A late glacial–early holocene multiproxy record from the eastern 912 fram strait, polar north atlantic. *Marine Geology* 355, 15-26 (2014).

Apart from that omission, I enjoyed reading the revised manuscript.

Response to reviewers' comment

The authors would like to thank the reviewer for their constructive comments. We have responded to each reviewer's comments below with our response in blue.

Reviewer #1 (Remarks to the Author):

This manuscript presents new data on the history of the NEGIS from the LGM through early Holocene. It builds on previous work that has shown that the ice margin reached the shelf edge at the Westwind Trough and that it had already retreat off the outer shelf during the LGM. What this manuscript adds is faunal evidence from a key core (144GC) and lithofacies analysis plus dating on two cores (144 and 146GC) to gain more detailed understanding of the further retreat history and what forcings were at play. The core sites target important ice marginal features (grounding zone wedges) on the mid/outer shelf. The lithofacies and foraminifera assemblage data are used to reconstruct important aspects of the NEGIS in the past: the configuration of the ice front (i.e. ice shelf fronted or ice cliff fronted); emissions of meltwater, advection of relatively warm RAC water, and sea-ice cover. I was struck by two key findings. The first is firm indication that the ice retreated during the LGM and during the oldest part of deglaciation, HS-1. The cores do not contain a complete record of this retreat as they were ice covered until c. 19 ka. A basal diamicton interpreted as subglacial deposit is overlain by fine grained laminated sediment lacking ice rafted clasts and containing a fauna consistent with an ice shelf fronted margin. The second key result to my mind is that the GZWs record relative stability of the grounding zone during the interval of c. 19ka to c. 15.8 ka. After c. 15.2 ka, the lithofacies indicate a calving margin lacking an ice shelf and the faunas indicate strong inflow of RAC and relative ocean warmth at the seabed that would influence ice instability. The GZWs had been thought to be Younger Dryas in age, so this is a key finding! The conclusion is that ice retreat from the GZW 2 may have been driven both by warm atmosphere and ocean forcing. I think that the data exist in this paper to make these points. However, there are some structural problems and some lack of clarity in the manuscript that hurt its ability to make these points clearly.

The Results section does not clearly distinguish previous work and background information from new results presented in this study. I would argue that the first 3 paragraphs are actually part of an introduction, setting the stage for the problems that the core data can attempt to solve. The fourth paragraph is meant to indicate how the lithofacies and foraminifera in the two cores can be associated with environments, but it was confusing because it goes immediately to interpretations without presenting the characteristics of the lithofacies and fauna and how these progress upcore and in association with the chronology. In a sense, there is no results section in this paper. The data are not described in a systematic way before interpretations are made. It is difficult to separate what is new and what is previously published.

We have made structural changes throughout the manuscript. We have moved and adapted the first three chapters of the results section into a new sub-section of the introduction titled

‘Background to study site’. We have incorporated the 4th paragraph into the section titled ‘Early deglacial and Atlantic water influence (>20.3 ka BP – 17.6 ka BP)’.

We have restructured each results sub-section so the results from this study are described first then interpreted. The paper now has a much clearer logical progression that we hope does justice to the findings of the paper.

Line 189 there is an error in the sedimentation rate calculation...I think it is off by two orders of magnitude. Maybe what is meant is 0.2 cm per year rather than per kyr. This is indeed incorrect. We have also modified the chronology using a new calibration method that has changed the original sedimentation rate. Instead, we have adapted the text to state ‘However, there is no clear evidence of a change in sedimentation rate that could reduce benthic foraminiferal productivity.’ (Lines 204-205)

The figures are nicely drawn, but they fall short of helping to convey the points being made in some cases.

On Figure 1, the map of the area, two GZWs are shown. Core 144 is shown immediately in front of the outer GZW. I was however, confused that on the acoustic profile of the core site in Figure 3, that core 144 is shown to the NW of GZW1. Is this a matter of the directions being misplaced on the figure? I am unsure, but think this profile needs to be turned around. It would help as well if Figure 1 showed where the two profiles are in the trough so that the orientation and locations are clear. In addition, it would help the introduction if there was a panel added to show the modern NEGIS, maybe using flow speed. Otherwise it is not easy for uninitiated readers to match the outlet glaciers with the main ice stream.

This was confusing so we have now rewritten and clarified it. The feature to the east of 144GC is not GZW1 and we confirm that the direction of the line in the original image is correct. Instead, this is another ice marginal feature that lies to the east of the core 144GC and closer to the shelf edge. It therefore formed prior to GZW1. Given its geometry and acoustic properties, we believe this to be an additional GZW (cf. Batchellor and Dowdeswell, 2015). It is now labelled it as an additional GZW on Figs. 2 and 6. Location of the two acoustic lines are marked on Fig1 in orange. We have added the Ice sheet velocity measurements to Fig 1a and labelled the location of NEGIS which is evidence by the section of fast flowing ice.

Figure 4 could benefit from having the Lithofacies along the depth axis as in Figure 2 so a reader does not need to flip back and forth. Suggest you include planktic forams per ml and benthic forams per ml. You could use the planks per ml instead of the P:B ratio if space is an issue. The ice species are in purple not blue...at least in the reviewer pdf. I presume that faunal analyses were not made on core 146 which is too bad.

We have added the lithofacies along the depth access as requested. We have removed P:B ratio. Both planktic and benthic foraminifera per ml is added to the figure. The colour of the ice species is supposed to be blue, but we have adopted the colour to a different blue colour so it doesn't appear purple on screens or printouts.

Figure 5. needs some additional annotation such as chrons (LGM, HS1, B-A, YD etc). Maybe add the modern average air temperature to the summer air temp panel. Might be useful as well to tie in the faunal zones and lithofacies as these are important as well as the fauna.

We have added both the lithofacies, including the IRD layers in core 144GC, as well as the faunal zones to the figure. We have annotated on the figure the timing of Greenland stadial 1 and 2, Greenland interstadial 1 and Holocene. We have not added modern air temperature to the panel as there is already a lot in this figure.

Figure 6 shows schematic profiles of the reconstructed NEGIS through time. Panel a indicates that the ice margin retreated by calving back and that dirty ice bergs were released. Early in the paper there is evidence of IRD in slope cores during this period. Panel b is the starting point of this study. But this panel does not match the data. GZW 1 is a bump on the seafloor but no association made with the core. Is GZW 1 associated with deposition of the laminated facies lacking IRD? Does that suggest that this is an ice shelf fronted margin beginning from the very start of the core record? This schematic does not explain GZW 1 and how it relates to results presented for 144GC. Panel C makes sense except that the story either is missing the IRD interval of ice retreat to GZW 1, or that the formation of GZW 1 is ignored. Presumably the GZWs are formed as the ice stabilizes, so perhaps they should not appear until the panel in which they form.

We have redrawn Figure 6 so that it aligns more closely to the timing of events as written in the text. We have removed GZWs in the panels where ice is grounded and only included them in the panels in which they are formed. We have included the new GZW that lies in front of core 144GC in panel B. Panels b and c now show the persistent presence of an ice shelf which account for the lack of IRD in panel C that the reviewer highlighted.

Reviewer #2 (Remarks to the Author):

Review of Callard et al (2025) - 'Ocean driven retreat of the Northeast Greenland Ice Stream following the Last Glacial Maximum'

Summary

This study presents data and conclusions from two new sediment cores taken on the Norske Trough that are used to describe the early deglaciation of the Northeast Greenland Ice Stream (NEGIS) from its LGM position. The paper has numerous interesting findings that are used to describe the evolution of the paleo-ice stream by tying in sediment facies, analysis of foraminifera assemblages and glacial landforms. Chief among them is the important role that ocean melt, driven by warm Return Atlantic Water Current (RAC), played in the initial retreat of NEGIS from its LGM by ~20 ka BP, a retreat that occurred without a rise in atmospheric temperature and that was much earlier than previously thought. RAC is further linked with

the collapse of the ice shelf after 15.7 ka BP when surface air temperatures began to rise during the onset of the Bølling-Allerød. Furthermore, Grounding Zone Wedges in the Norske Trough provided vital pinning points and stability during the early deglaciation after the trough deepens into a retrograde slope inland of the shelf edge.

I found this paper very enjoyable to read and believe it shares valuable insights into the evolution of the NEGIS from the LGM. This is especially topical as constraining the future response of the contemporary NEGIS to climate warming is crucial. The manuscript is well-written and well-structured, with the authors clearly explaining their interpretations of sediment record which I, for the most part, agree with.

My comments below mainly relate to interpretations made around the early retreat/deglaciation of the NEGIS, specifically whether an ice shelf existed before 18 ka BP, and the role the bathymetry and fjord topography played in the initial retreat. I will state here that my opinions are far from set in stone, and hopefully adding a few sentences to further explain their interpretations would alleviate any of my concerns and lead me to recommend this paper for publication.

Major Comments

- **Ice Shelf Interpretation:** I agree with the authors' interpretation that an ice-shelf existed in the Norske Trough with the grounding line on GZW2 between 18-15ka BP. However, I'm curious why the authors believe an ice shelf wasn't present before NEGIS retreat to GZW2 (for example in Figure 6B). Instead, during the early deglaciation (>20 ka BP to 18 ka BP), the high levels of *S. horvathii* and lack of IRD are presented as reasons for thick sea-ice conditions. These same arguments are then used as evidence for an ice shelf between 18-15 ka BP. Furthermore, as the authors rightly mention, the occurrence of GZWs are indications of shelf ice conditions, therefore I would expect it likely that an ice shelf was present with the grounding line on GZW1 during the early deglaciation (López-Quirós et al., 2024). I also wonder why the keel marks from tabular ice bergs must have been formed after 18 ka BP, could they not have been formed from an earlier ice shelf?

Reviewer 2 is correct, and we have now clarified and amended this aspect of the paper. We agree with the reviewer that the core lithology and foraminiferal evidence point to the presence of an ice shelf throughout the whole early deglacial period until 15.2 ka BP. This is clearly supported by the geomorphic evidence (GZW's) on the sea floor. An extensive iceshelf was over both core 146 and 144 prior to 15.2 ka, when it subsequently broke up leading to the IRD layers of the same age recorded in both cores. We have adjusted the wording throughout the manuscript to reinforce this message. It is the most critical aspect of this story. Changes in the foraminiferal assemblage and flux data indicate a change in the distance of the ice-shelf front to the core site and we have added this detail to the text. In our initial submission we also considered the likelihood of sea-ice cover as an alternative to the presence of an ice shelf, but this was a weaker argument and overly complicated. It has now been de-emphasised.

Later in the discussion it is stated ‘... that NEGIS was fronted by an ice shelf when it stabilised on GZW2’. This is not what it shown in Figure 6B when the retreat is shown as a calving cliff face with harsh sea-ice conditions when it reached GZW2. It is then said that the presence of an ice shelf was of ‘negligible’ importance to the initial stability of NEIGS and that the bathymetric high was more important. I think that a reference or two is required here, as I’m not entirely convinced by that conclusion.

We thank the review for their comment and have addressed this in the text to state that the ice-shelf acted to stabilise the retreating NEGIS. We have included a reference to the modern ice-shelves from the area to support this. However, the location of the bathymetric high is also an important control on stability of the ice stream as well as the ice shelf.

- Bathymetry and Fjord Geometry: I agree with the authors that RAC was likely the cause of early retreat and echo that this is a great finding of the paper. However, I’m not sure I fully agree with the authors comments on the fjord geometry and slope angle regarding this initial retreat, where the initial slow retreat rates (23 to 19 m/yr) on a retrograde slope are used to conclude the bathymetry was not a principal control. Firstly, I think it would be great to have a comparison against retreat rates from other paleo-ice streams in Greenland, is 23 m/yr considerably slower? Secondly, I see that the Norske Trough immediately narrows from the shelf edge, could this have slowed initial retreat through increased lateral drag, similarly as the authors later discuss with the narrowing of the trough around GZW2. Finally, while the bathymetry deepens from 320 m at the shelf edge to 560 m 100 km in land, the majority of this deepening (210 m) happens ~50km from the shelf edge, therefore, wouldn’t it be reasonable to while retrograde in nature, the initial shallow slope may have contributed to the slower retreat rates?

We have added two examples. During the early deglaciation of the Ummannaq Ice Stream that has retreat rates were 121 to 94 ma^{-1} . During the early deglaciation of Jakobshavn Isbrae retreat rates were 104 to 137 ma^{-1} (Roberts et al. 2024). Both are order of magnitude higher than the rates reported in this paper and highlights that the NEGIS retreat was comparatively slow. We agree that lateral drag could influence the retreat rate. We have added additional wording to this effect in the manuscript.

Minor Comments

- Figure 1: I think the polygons for GZW1 and 2 should only have the outlines and not be filled, it would then be possible to see how the GZWs look in comparison to the sea floor. I also wonder is the scale for the bathymetry could be clipped to range from 0 to 1500 m, just to make it the variations in depth slightly clearer.

The polygons have been changed to outlines and we have clipped the bathymetry to 0-1500.

- Figure 5: JJA air temperatures. I think this should be made clear is if for the contemporary NEGIS region or the paleo domain of the ice stream? I’m also curious if the temperatures are any different in the newer Badgeley et al., 2020 dataset?

The modelled data is for a specific location rather than the whole NEGIS region. We have added some extra detail in the caption. We have not made a comparison between temperature reconstructions as we are interested in timing and pattern of change as a clue to possible drivers opposed to absolute temperature changes.

- 1.34: It would be worthwhile mentioning that 79N still has an ice tongue that is providing significant buttressing to the interior ice (Humbert et al., 2023), especially as you discuss ice shelf evolution and collapse in the results/discussion.

We have added the following text '79N still retains its 70 km long, 20 km wide ice shelf, that buttresses the ice stream. However, a significant calving event observed in 2022 indicates ongoing ice-shelf instability, with models predicting that collapse of the ice shelf would increase grounding line flux by over 160%⁶.' (lines 38-41)

- 1.24: ice shelf break 'up'? This is corrected
- 1.29: Redefine the Greenland Ice Sheet acronym - GrIS. This is redefined in line 32
- 1.36: Capital W for 'with'. Corrected
- 1.40 – 42: Sentence is a bit sloppy. 'There' should be 'their', and maybe reads better as '... their reliability depends on an accurate parametrisation of ice-sheet physics, including key boundary conditions such as...'

We have changed the wording to 'While numerical simulations provide a powerful predictive tool, their reliability depends on an accurate parameterisation of ice-sheet dynamics and key boundary conditions such as bathymetry and climate forcing.' (lines 48-50)

- 1.45: 'enabling a better understanding of ice-ocean interactions' – We already have some understanding of how ice-ocean, but the paleo record definitely improves this.
- 1.51: double 'from' – maybe better as 'Recent evidence from marine cores taken on the continental shelf...' We have changed the wording to 'Recent evidence from marine cores, bathymetric and geophysical observations from the continental shelf provides a longer-term reconstruction of the behaviour of the NEGIS.' (lines 68-69)
- 1.57: '... initial retreat of NEGIS and critically...' Corrected
- 1.94: I am presuming that the moraine ridge referred to here as not GZW1? I think this could be made a little clearer by also stating that 144GC is also in front of GZW1. We have adapted the text and figures so that this is clearer. (lines 100-101: 'Core PS100-144GC (hereafter 144GC) lies between two GZW's (a newly identified GZW and GZW1; Fig. 1 and Fig. 2).')

- L.138: Missing full stop. Corrected
- 1.146 where is core CR02-GC? I don't see it in Figure 1. We have added GR02-GC to Figure 1.
- 1.149: Incorrect capitalisation of D. Corrected

- 1.164: Full stop within the brackets when it should be outside. Corrected
- 1.180: Add reference to Fig.4 at the end of the sentence. We have changed the wording to demonstrate how low the foraminiferal counts were in the shaded sections.
- 1.258: Missing bracket Corrected
- 1.259: ‘Driven melting’ and ‘driven ice shelf breakup’, choice of words could be improved in this sentence. This section has been rephrased ‘The timing of ice-shelf breakup, ~15.2-14 ka BP, corresponds with an increased RAW signal on the continental shelf (increase in *C. neoteretis* shown in 144GC) suggesting that ocean-driven melting played an important role in ice-shelf break up.’ (lines 314-316)
- 1.282: Missing full stop. Corrected
- 1.287: Missing full stop. Corrected
- L.291: 23 m/yr is for the Norske Trough while 19 m/yr is the Westwind trough in reference 20. This should be made clearer (Ó Cofaigh et al., 2025). We have removed reference to the 19ma⁻¹ in Westwind Trough (Lines 348-350)
- 1.334: ‘melting was an important driver’ Corrected
- 1.353: Should AWI not be AW? Corrected
- 1.353: Missing spaces and doubles spaces in this sentence. Corrected

References

- Badgeley, J.A., Steig, E.J., Hakim, G.J., Fudge, T.J., 2020. Greenland temperature and precipitation over the last 20000 years using data assimilation. *Clim. Past* 16, 1325–1346. <https://doi.org/10.5194/cp-16-1325-2020>
- Humbert, A., Helm, V., Neckel, N., Zeising, O., Rückamp, M., Khan, S.A., Loebel, E., Brauchle, J., Stebner, K., Gross, D., Sondershaus, R., Müller, R., 2023. Precursor of disintegration of Greenland’s largest floating ice tongue. *The Cryosphere* 17, 2851–2870. <https://doi.org/10.5194/tc-17-2851-2023>
- López-Quirós, A., Junna, T., Davies, J., Andresen, K.J., Nielsen, T., Haghypour, N., Wacker, L., Olsen Alstrup, A.K., Munk, O.L., Rasmussen, T.L., Pearce, C., Seidenkrantz, M.-S., 2024. Retreat patterns and dynamics of the former Norske Trough ice stream (NE Greenland): An integrated geomorphological and sedimentological approach. *Quat. Sci. Rev.* 325, 108477. <https://doi.org/10.1016/j.quascirev.2023.108477>
- Ó Cofaigh, C., Lloyd, J.M., Callard, S.L., Gebhardt, C., Streuff, K.T., Dorschel, B., Smith, J.A., Lane, T.P., Jamieson, S.S.R., Kanzow, T., Roberts, D.H., 2025. Shelf-edge glaciation offshore of northeast Greenland during the last glacial maximum and timing of initial ice-sheet retreat. *Quat. Sci. Rev.* 359, 109326. <https://doi.org/10.1016/j.quascirev.2025.109326>

Reviewer #3 (Remarks to the Author):

Callard et al., provide new information relating to the role oceans played in governing the retreat of the Northeast Greenland Ice Sheet on the mid/outer continental shelf. Key results are:

- Deglaciation began at around 20ka BP on the mid/outer Northeast continental shelf, this is earlier than previously reported in the Norske Trough.
- Analysis of foraminiferal assemblages suggests this was primarily driven by the intrusion of warm Return Atlantic Water onto the shelf, amplified by a retrograde seabed.

Overall, I liked that paper, and I found it interesting; the discussion was enjoyable to read. I believe that the contribution is important in this region, particularly given the impact of warming ocean waters on the fate of the Greenland Ice Sheet today and in the future. I think that the methodologies and science are sound, I have some comments on this in the 'main comments' section of this review.

One of the main comments I have relates to the quality of the writing and mistakes throughout. I believe that these detract from the overall manuscript. I therefore list those small changes first here. However, I recommend that the entire manuscript be carefully proofread edited prior to subsequent review.

We have worked hard to improve the quality of writing and remove mistakes.

Comments relating to quality of writing and figures

- Sometimes the authors use 'the' NEGIS sometimes 'the' is excluded – it should be consistent We have added the to all occurrences of NEGIS.
- Type in figure 1a caption – 'value' coldest, I think this should be blue. We have corrected this to blue
- Line 41 – their reliability not there reliability Corrected
- RAC never defined. I think it would be more correct to use Return Atlantic Water (RAW) to describe the water mass in some instances. E.g. like 309. This should be checked throughout We have changed all RAC to RAW in the text and figures for consistency.
- The full core name of 135G should be given in the figure caption of 1b, as per the other cores listed Full core name has been added in caption

- What is AWI Bank – not defined anywhere. This is a name for the Shoal that runs on the southeastern edge of inner Westwind Trough as published in Arndt et al 2015 and 2017. This is not defined but assumed to be Alfred Wegner Institute bank.
- The Rasmussen et al., 2022 core should be marked on the map (19G), it is in the caption but not on the map. 39G is also listed in the caption but not on the map. Please check all of these again. We have added 19G to the map. Mention of 39G has been removed from the caption as this core is not referred to in the main text
- I think fig 2 and 3 should be switched as fig 3 is introduced in the text first and this is confusing We have switched the figures
- Fig 4: foraminifera species names should always be in italics, also in Fig 5 This has been corrected
- Fig 4 caption, should this not be <300 counts Yes, the reviewer is correct, but we have also changed the wording at the end of this caption.
- Please check the spelling of all species throughout the manuscript. I noticed some mistakes (e.g. s. hovathi has two I's on line 141) We have checked the spelling of all species names throughout the text and corrected those that were misspelled.
- No references linked to *S. feylingi* inferences – it is my understanding that this species is linked to productivity (e.g. Seidenkrantz, 2013). In the original text we used reference 38 Knudsen, K. L. & Seidenkrantz, M.-S., 1994. *Stainforthia feylingi* new species from arctic to subarctic environments, previously recorded as *Stainforthia schreibersiana* (Czjzek). In H.P. Sejrup, K.L. Knudsen (Eds.), *Late Cenozoic Benthic Foraminifera: Taxonomy, Ecology, and Stratigraphy*. Cushman Foundation for Foraminiferal Research, Special Publication, vol. 32 (1994), pp. 5-13.

We have since added Patterson, R. T., et al. (2000). "Oxygen level control on foraminiferal distribution in Effingham inlet, Vancouver Island, British Columbia, Canada." *The Journal of Foraminiferal Research* 30(4): 321-335.
- Typo line 149 – capital D corrected
- Space missing line 258 after 144GC corrected
- Typo line 259, remove driven corrected
- Full stop missing from line 282 corrected
- Full stop missing line 287 corrected
- I would start a new paragraph for “as previously.” (line 320). corrected
- Foraminifera should be foraminiferal when it is used as an adjective (e.g. foraminiferal assemblages, line 323 and foraminiferal species, line 409). This should be checked throughout the manuscript and changed accordingly. We have checked and corrected all occurrences where this change is required.

- References required for ice shelf statement – line 328 We have included reference 53: Gudmundsson, G. Ice-shelf buttressing and the stability of marine ice sheets. *The Cryosphere* 7, 647-655 (2013).
- What is AWI (Line 353) This is corrected to RAC
- Indicate not indicates (line 353) corrected
- Space after 270GC (Line 254) corrected
- Indicate not indicated (Line 358) corrected
- Constraints not constrains (Line 360) corrected
- Extra l before planktic in fig caption 5a corrected
- I think it would be better to be consistent using line graphs or bar charts for the foraminifera data in Fig 4 and 5 After making comparison figures we have decided to keep the different styles between figures as it made the figures clearer. Similar approaches have been used in other published articles (e.g., Lloyd et al, 2023)
- The end of Figure 6 caption needs removing. And the correct description of the figure included. An appropriate figure caption has been added: ‘Figure 6: Conceptual model representing the interaction between the marginal position of the NEGIS, ocean circulation and ice shelf activity. Panels a) to h) represent key time periods discussed in the manuscript and consists of a cross-sectional view of the NEGIS and the fronting ice shelf as well as a top-down view showing extent in relation to the current NEGIS continental shelf. Bathymetry, position of GZWs and position of sediment cores analysed are illustrated and are estimations.’

Main comments

- I very much enjoyed reading the discussion, I think the argument for RAC is strong with the nuances discussed and previous evidence well incorporated! Thank you for this comment.
- There is a wealth of recent literature highlighting past changes to the NEGIS, for me, the fact that deglaciation began earlier than previously reported needs to be stated in the abstract as I think it is a very important finding and differentiates this paper from those previously published. This is clear in your summary at the end of the paper (Line 360 start), but could be better stated in the abstract. Simply adding ‘Radiocarbon dating shows deglaciation began ~20 ka BP, earlier than previously reported, etc’ would suffice. We have edited the abstract, but it is important to note that O Cofaigh et al 2025 have published the oldest age for deglaciation (21.6 ka BP). However, this record provides the oldest reconstruction of RAW and confirms the continued retreat of the NEGIS back 100 km from the continental shelf edge to the core location and during the global LGM and we can link that retreat to RAW. We have adapted the abstract and text, so this is clearly stated.
- I think it is important to present the foraminiferal data as concentrations, (e.g. no. taxa/g dry sed), given the impact of compaction downcore. As volumetric data associated with the samples (as listed in the methods) was taken, I think it think this is possible if bulk sediment

weights are/were made. If this is not possible, I think foraminifera fluxes could be useful to include. Sample were volumetrically measured not weighted and therefore we cannot calculate the taxa/g dry weight. However, we have calculated the foraminiferal fluxes and referred to this throughout the results section and added the data to Fig 4 and 5.

- I think that this paper should be cited, given that it examines the distribution of modern benthic foraminifera in the exact study area being investigated here and the proxies used:

o Davies, J., Lloyd, J., Pearce, C. and Seidenkrantz, M.S., 2023. Distribution of modern benthic foraminiferal assemblages across the Northeast Greenland continental shelf. *Marine Micropaleontology*, 184, p.102273. Useful inferences about specific species should be added from this (e.g. *Stetsonia horvathi*)

Thank you for highlighting this omission. We have referenced this paper where appropriate throughout the manuscript.

- Fig. 2. I do not understand the reasoning behind the order of the xray images in the red box. I think it make more sense to list this in order of appearance in the cores. It would also be helpful if the lithofacies abbreviations were defined in the caption. The order of the lithofacies have been changed in the figure so they are in order of appearance.

- I think this reference is also important to include: Tabone, I., Robinson, A., Montoya, M. and Alvarez-Solas, J., 2024. Holocene thinning in central Greenland controlled by the Northeast Greenland Ice Stream. *Nature communications*, 15(1), p.6434. It provides useful information relating to the importance of constraining timing and speed of retreat of the NEGIS and I believe would be useful for the introduction. This is indeed a useful paper to cite, and we have included reference to this paper (reference 28) in the introduction (Lines 92-96)

- To me it seems like these cores are located on the mid/outer shelf, I wouldn't describe 146GC as being on the outer continental shelf (Line 71). We have changed the text to outer continental shelf. (Lines 109-110)

- Given the high abundance of *C. reniforme*, I think more focus needs to be given to its interpretation in the results and discussion. Broadly, it seems to covary with *C. neoteretis* and previous studies have linked it to chilled Atlantic water. I think this should be included and discussed. We agree and have added more detailed about *C. reniforme* and its covariance with *C. neoteretis* as well as it's link to chilled Atlantic Water.

- I would find it helpful to have the Faunal zones marked on Fig 5. These have been added to the figure

- Why is the age model for 146GC not included (e.g. as part of supplementary figure 3)? We did not conduct an age model for core 146GC due to the limited chronological constraint, particular for the older section of the core. We have added a line in the methods explaining this.

- Were all benthic foraminifera species included in the radiocarbon samples? Or were some species (e.g. *Miliolina*) excluded. This is important to outline these details in the methodology. It was mixed but we did avoid *Miliolina* and *Quinqueloculina* species, I have

added this detail in the methods.

References:

Arndt, J.E., Jokat, W., Dorschel, B., Myklebust, R., Dowdeswell, J.A. and Evans, J., 2015. A new bathymetry of the Northeast Greenland continental shelf: Constraints on glacial and other processes. *Geochemistry, Geophysics, Geosystems*, 16(10), pp.3733-3753.

Arndt, J.E., Jokat, W. and Dorschel, B., 2017. The last glaciation and deglaciation of the Northeast Greenland continental shelf revealed by hydro-acoustic data. *Quaternary Science Reviews*, 160, pp.45-56.

Lloyd, J. *et al.* Ice-ocean interactions at the Northeast Greenland Ice stream (NEGIS) over the past 11,000 years. *Quaternary Science Reviews* **308**, 108068 (2023).

Ó Cofaigh, C., Lloyd, J.M., Callard, S.L., Gebhardt, C., Streuff, K.T., Dorschel, B., Smith, J.A., Lane, T.P., Jamieson, S.S.R., Kanzow, T., Roberts, D.H., 2025. Shelf-edge glaciation offshore of northeast Greenland during the last glacial maximum and timing of initial ice-sheet retreat. *Quat. Sci. Rev.* 359, 109326. <https://doi.org/10.1016/j.quascirev.2025.109326>

Response to reviewers' comment

The authors would like to again thank the reviewer for their comments. We have responded to each reviewer's comments below with our response in blue.

REVIEWERS' COMMENTS

Reviewer #1 (Remarks to the Author):

The revised manuscript is much better and I do not have any additional comments or suggestions. I think it is well written and illustrated and the conclusions are noteworthy.

Reviewer #2 (Remarks to the Author):

2nd Review of Callard et al (2025) - 'Ocean driven retreat of the Northeast Greenland Ice Stream following the Last Glacial Maximum'

Summary:

The authors have done an excellent job in responding to the comments from myself and the other reviews. This new version of the manuscript is both an improvement in structure and writing; while also further emphasising, valuable insights gained from the paleo-evolution of NEGIS and crucially the importance of ocean melt in the stability of ice shelves and grounding line behaviour.

My previous comments mainly concerned the interpretations of glacial dynamics at the beginning of the deglaciation, whether an ice shelf existed during retreat from the shelf edge, as well as the role of bathymetry in driving retreat. Both concerns have been alleviated through added references and greater context in the discussion, as well as improvements to Figures 1 and 6. Furthermore, there are numerous small changes throughout the manuscript that address all my minor comments completely. I would also like highlight the restructuring of the manuscript to include the 'Background to study site' section after the Introduction. This was in response to Reviewer #1, and I believe this greatly improves the readability of the manuscript and enhances the story of NEGIS retreat of the LGM and the importance of warm Atlantic water.

To conclude, I believe this piece of work provides novel and crucial insight into the deglaciation of the Greenland Ice Sheet from the LGM and importantly the role of warm Atlantic water is driving grounding line retreat and ice shelf instability. The early retreat of the NEGIS (~20 ka BP) despite a lack of atmospheric warming, plus the enhanced Atlantic water signal around the collapse of the ice shelf are vital conclusions that stress the importance of the ice-ocean boundary as we strive to understand the retreat of contemporary ice sheets through both observations and modelling efforts.

I just have a few minor comments relating to grammar:

- In the response to reviewers there is mention of a “Background to Study Site” section, however I can’t see this sub-heading. Maybe an issue with the PDF but I want to highlight in case. This has been added, thank you.
- Line 31-32: Maybe could be re-written to avoid saying the Greenland Ice Sheet twice in quick succession. This has been reworded to: The Northeast Greenland Ice Stream (NEGIS) extends more than 600 km into the interior and drains approximately 12% of the Greenland Ice Sheet (GrIS).
- Line 195: Missing space between cores and 114GC : added
- Line 217: Comma should be a full stop corrected
- Line 349: Missing space after ‘approximately’ corrected
- Line 357: Missing space after GZW corrected
- Line 357: Miss placed bracket after “moraine ridges” corrected
- Line 364: Missing bracket and full stop after Fig 5. corrected
- Line 393: Missing bracket before Figs corrected

Jamie Barnett
Stockholm University

Reviewer #3 (Remarks to the Author):

I think the author's have adequately addressed the comments made on their manuscript. However, I could not see this reference in the reference list:

References required for ice shelf statement – line 328

We have included reference 53: Gudmundsson, G. Ice-shelf buttressing and the stability of marine ice sheets. *The Cryosphere* 7, 647-655 (2013).

I could not see this reference in the text. Reference 53 refers to Aagaard-Sørensen, S. et al. A late glacial–early holocene multiproxy record from the eastern 912 fram strait, polar north atlantic. *Marine Geology* 355, 15-26 (2014). This paper has been referenced and is number 53. Aagaard-Sorensen is number 54. (line 678)

Apart from that omission, I enjoyed reading the revised manuscript.